# Structures of insect Imp-L2 suggest an alternative strategy for regulating the bioavailability of insulin-like hormones

Nikolaj Kulahin Roed [1], Cristina M. Viola[2], Ole Kristensen [3], Gerd Schluckebier [1], Mathias Norrman[1], Waseem Sajid[1], John D. Wade [4,5], Asser Sloth Andersen [1], Claus Kristensen [6], Timothy R. Ganderton[2], Johan P. Turkenburg [2], Pierre De Meyts [1,7] & Andrzej M. Brzozowski [2]

The insulin/insulin-like growth factor signalling axis is an evolutionary ancient and highly conserved hormonal system involved in the regulation of metabolism, growth and lifespan in animals. Human insulin is stored in the pancreas, while insulin-like growth factor-1 (IGF-1) is maintained in blood in complexes with IGF-binding proteins (IGFBP1–6). Insect insulin-like polypeptide binding proteins (IBPs) have been considered as IGFBP-like structural and functional homologues. Here, we report structures of the *Drosophila* IBP Imp-L2 in its free form and bound to *Drosophila* insulin-like peptide 5 and human IGF-1. Imp-L2 contains two immunoglobulin-like fold domains and its architecture is unrelated to human IGFBPs, suggesting a distinct strategy for bioavailability regulation of insulin-like hormones. Similar hormone binding modes may exist in other insect vectors, as the IBP sequences are highly conserved. Therefore, these findings may open research routes towards a rational interference of transmission of diseases such as malaria, dengue and yellow fevers.

[1] Global Research, Novo Nordisk A/S, Novo Nordisk Park 1, 2760 Maaloev, Denmark. [2] York Structural Biology Laboratory, Department of Chemistry, The University of York, Heslington, York YO10 5DD, UK. [3] Department of Drug Design and Pharmacology, University of Copenhagen, DK-2100 Copenhagen Ø, Denmark. [4] Florey Institute of Neuroscience & Mental Health, University of Melbourne, Parkville, VIC 3010, Australia. [5] School of Chemistry, University of Melbourne, Parkville, VIC 3010, Australia. [6] Department of Cellular and Molecular Medicine, University of Copenhagen, Blegdamsvej 3, DK-2100 Copenhagen N, Denmark. [7] Department of Cell Signalling, de Duve Institute, B-1200 Brussels, Belgium. Correspondence and requests for materials should be addressed to A.M.B. (email: marek.brzozowski@york.ac.uk)

The insulin/insulin-like growth factor system is an evolutionarily ancient, highly conserved, endocrine and paracrine signal transduction network in multicellular organisms[1]. It regulates a broad spectrum of key life processes such as organism metabolism, somatic growth, development, reproduction and lifespan, and cell growth, differentiation and migration at the cellular level. In humans and many vertebrates, insulin is stored in oligomeric forms in pancreatic β-cells, and is acutely secreted in response to glucose and nutrients[2]. It circulates freely in the blood after secretion with a short (~4 min) half-life[3]. In contrast, human insulin-like growth factors (hIGF-1 and 2) are secreted by multiple tissues and do not oligomerise but are tightly bound in biological fluids by several IGF binding proteins (IGFBP1–6)[4–6]. Ultimately, insulin and IGFs exert their signalling through closely related tyrosine kinase type insulin and IGF-1 receptors (IR, IGF-1R)[1,7,8](Fig. 1).

The occurrence of similar insulin/IGFs-like hormones (insulin-like proteins, ILPs), is very diverse in the animal kingdom, ranging from one insulin and IGF-1/IGF-2 in human, eight ILPs in *Drosophila* (DILP1–8), to 40 ILPs in *C. elegans*[1,7,9–13]. Nevertheless, they share similar motifs of inter-/intra-chain disulphide bridges and organisation into B, C, A and D-domains, typical of insulin, pro-insulin and IGF-1/2 (Fig. 2, Supplementary Figure 1). The variety of types of ILPs is not reflected in the number of their receptors that is limited to one insulin receptor (IR, with IR-A and IR-B isoforms) and one IGF receptor (IGF-1R) in humans, to one IR-like in most invertebrates, including insects such as *Drosophila* (dmIR)[1,7].

The intracellular ILPs-signalling pathways—so called insulin/IGF axis—are well conserved in the animal kingdom[14], in contrast to the diverse regulations of bioavailability of these hormones. In humans, this involves six blood-circulating IGFBPs and the non-signalling IGF-2/mannose-6-phosphate receptor (IGF-2R)[15–20] (Fig. 1). The ~213–289 amino acids long, three-domains human IGFBP-1-6 tightly regulate the level of free IGF-1/2 by forming binary/ternary complexes with these hormones[4–6,16,17,19–21].

An understanding of IGFBP-1–6 was further convoluted by a proposed extension of the IGFBP superfamily by a number of cysteine-rich proteins[17], which homology to IGFBP1-6 is limited to their N-terminal IGFBP-like domain, with suggested IGFs affinities ~100× lower than of the true IGFBPs. These proteins were termed as IGFBP-related proteins (IGFBP-rPs)[17], and they

included Mac25 (temporarily renamed also as IGFBP-7), which cDNA was originally identified in meningioma[22,23].

Subsequently, it was reported that the cells from the Fall armyworm, *Spodoptera frugiperda*, used in baculovirus Mac25 expression secrete a binding protein, named *Sf* ILPs binding protein (*Sf*-IBP), that can bind human insulin, IGF-1 and IGF-2[24,25], with high nM affinities, and that it can interfere with the formation of human insulin:IR complex and its signalling[25]. These and other findings[26] revealed that the postulated IGF-binding properties of Mac25 actually resulted from the contamination of the expression media by *Spodoptera* endogenous *Sf*-IBP, being confused further by similar ~27 kDa molecular masses of these proteins. As the other members of the purported IGFBP-rPs family also failed to show any insulin/IGFs binding[27,28], there is now a common consensus that only the six classical, human IGFBPs-1–6 should be designated as IGFBPs[5], and that *Sf*-IBP may represent a distinct ILP-bioavailability regulating system (Fig. 1),

The cloning of *Sf*-IBP indicated its high homology to the 242 amino acid *Drosophila* imaginal morphogenesis protein-late 2 protein (Imp-L2)[25] that is expressed during imaginal discs morphogenesis[29]. The secretion of Imp-L2 is induced by 20-hydroxyecdysone, and is implicated in neural and ectodermal development in *Drosophila*[30], with Imp-L2 null progeny from Imp-L2 null mothers, showing 100% lethality[30]. Imp-L2 can bind human IGF-1, IGF-2, insulin, and DILP5 with nM affinities[25], it counteracts insulin signalling in *Drosophila* being essential for resistance to starvation, while its overexpression leads to an extension of the lifespan[31,32].

Therefore the aim of this work was to provide the structural insight into the invertebrate ILPs-regulatory IBPs-based system, and to assess its molecular relationship to human IGFBPs. Here, we report the crystal structures of the *Drosophila* Imp-L2 protein in its apo-form, and in holo-complexes with insect DILP5 and human IGF-1. They are supported by ITC and surface plasmon resonance (SPR) Imp-L2 binding data with human insulin, IGF-1, and DILP5, and size exclusion chromatography with multi-angle light scattering (SEC-MALS) and small-angle X-ray scattering (SAXS) in solution studies of oligomeric states of this protein.

The three-dimensional characterisation of the Imp-L2 IBP in its free and hormone-complex forms revealed its fold and hormone-binding mode that are entirely different from, and

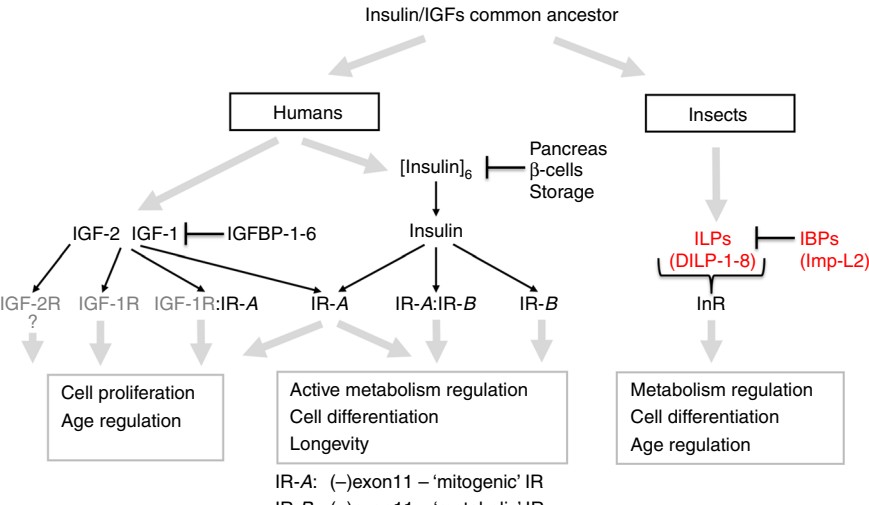

**Fig. 1** The main molecular steps in the insulin/IGF axes. IR-A, IR-B isoforms of the insulin receptor (IR), IGF-1R IGF receptor, and their heterodimers, IGF-2R IGF-2 receptor, IGFBP IGF-binding proteins, IBP ILP-binding proteins, [insulin]₆ crystalline, or other oligomeric, storage form of insulin in pancreas β-cells, DILP-1-8 and Imp-L2 *Drosophila* ILPs and IBP, respectively

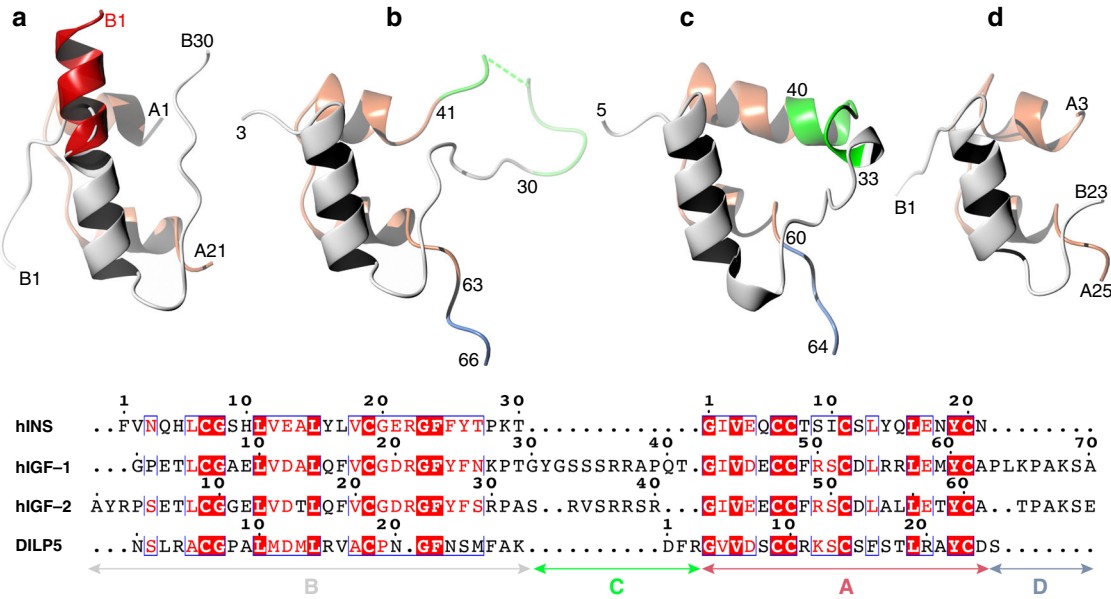

**Fig. 2** Structural organisation of representative insulin-like proteins. **a** human insulin (hINS in the sequence box)(PDB ID: 1mso), **b** human IGF-1 (1gzr), **c** human IGF-2 (hormone ligand taken from 3kr3 complex), **d** *D. melanogaster* DILP5 (2wfv). A-chains or domains (in IGF1/IGF2) are in coral, B-chains/domains in white, C-and D-domains in IGF-1/IGF-2 in green and blue, respectively (untraced residues in IGF-1 in green dashed line). The white insulin B-chain in **a** shows its T-conformation, while the red part of the B-helix depicts the R-conformer of the B1-B8 part of insulin B-chain

unrelated to, IGFBP-1-6. Hence Imp-L2 and, likely, other insects IBPs, represent an alternative macromolecular system for the control of the bioavailability of insulin-like hormones.

## Results

**Structure of apo-Imp-L2.** The apo-Imp-L2 consists of two, similar, immunoglobulin-(Ig)-like fold domains, each of them with two-sandwiched layers of ~6–7 anti-parallel β-strands. They contribute to an extensive, continuous β-sheet on the top-side of the Imp-L2 that is referred here as to inter-domain-β-sheet (id-β-sheet) (Fig. 3a). The N-terminal Ig-domain (1–142, referred here as to Ig-NT) is fused by a short linker to the C-terminal (145–242) domain (Ig-CT) (Fig. 3a). Although both domains can be classified as variants of the intermediate (I-set) subfamily of the Ig-fold they show significant variations from this structural motif, which likely reflects its adaptation to a novel organisation and function of the Imp-L2 (Fig. 3b).

The β-strands of Ig-NT domain consist of two β-sheets: βA′-βG-βF-βC and βA-βB-βE which generally follow the I-set fold (Fig. 3a, b)[33], but the I-set Ig typical βC′- and βD-strands are missing in the Ig-NT. Instead, their corresponding polypeptide chains form here a long, Imp-L2-characteristic 70–92 loop that joins directly the βC-strand and βE-strand (Fig. 3a, b). Moreover, a part of its putative βD-strand is fused with the βE-strand giving it an extra length that is needed for an effective formation of the Ig-NT:Ig-CT inter-domain interface. This departure from the classical I-set fold re-directs the 70–92 loop, which joins βE strand from a different/opposite direction than seen in Ig I-set domains (Fig. 3b).

The Ig-CT domain follows the Ig-I fold more closely, with its βC′-strand and βD-strand being present here. The short βA/βA′-strands of Ig-I are fused here into one, long βA-strand which together with the antiparallel βE-strand of the Ig-NT domain, forms a tight Ig-NT:Ig-CT inter-domain interface (Fig. 3a). However, this Ig I-set typical split of the βA/βA′ strands is reflected in the Ig-CT domain by a significant twist of this long (147–160) βA-strand. As a result, the N-terminal part of this βA

strand (147–154: referred her as to βA^N) belongs to βA^N-βB/βB′-βE-βD β-sheet of the Ig-CT, while the C-terminal part of this strand (157–160: referred here as to βA^C) contributes to the βA^C-βG-βF-βC-βC′ β-sheet of this domain (Fig. 3a, b). The structural strain of the Ig-CT βA-strand that is probably induced by its domain-interface function seems to be propagated into the neighbouring βB-strand of this domain, which is split into two βB (166–168) and βB′ (170–175) strands; a feature not observed in the Ig I-set. The distal, very short βC′-strand (187–189) is at a sharp (>50°) angle to the βA^C-βG-βF-βC plane and contributes poorly to this β-sheet.

The continuous inter-domain-β-sheet (id-β-sheet, top-side of the Imp-L2 (Fig. 3c) is formed by a tight interface between Ig-NT βA-βB-βE and Ig-CT βA^C-βG-βF-βC-βC′ β-sheets that is stabilised by hydrogen bonds between parts of Ig-NT βE (92–99) and Ig-CT βA^C (157–160) strands. The relative flatness of the id-β-sheet surface contrasts with a concave shape of the opposite (back) side of the Imp-L2, with its only one β−strand thick apex at the βA: βE inter-domain interface (Fig. 3c).

Despite some structural differences, the overall folds of the Ig-NT- and CT-domains are similar with an rms difference of 1.45 Å between their Cα chains, calculated after deletion of the 24–31, 72–93, 133–143 loops in the Ig-NT domain (referred here as to a core Ig-NT^Δ). A unique mirror-like edge-to-edge arrangement of the Impl-L2 Ig-domains brings close (~4.20 Å) its N-terminal and C-terminal, which is different than in the classical Ig-proteins were the N/C-termini are at the opposite ends of the domains, and which enhances further the planarity of the id-β-sheet.

**Relation of Imp-L2 to Ig-fold containing proteins.** The overall structure of the Imp-L2 protein does not have any obvious structural homologues in the Protein Data Bank (PDB). However, the Ig-NT and Ig-CT domains are remarkably similar to the M10 domain (M10^1–99, Supplementary Figure 2) that is the most C-terminal Ig-I-like subfamily segment of the human giant muscle protein titin[34–37], which is a docking platform for several sarcomeric binding partners such as obscurin[38]. The Ig-CT domain

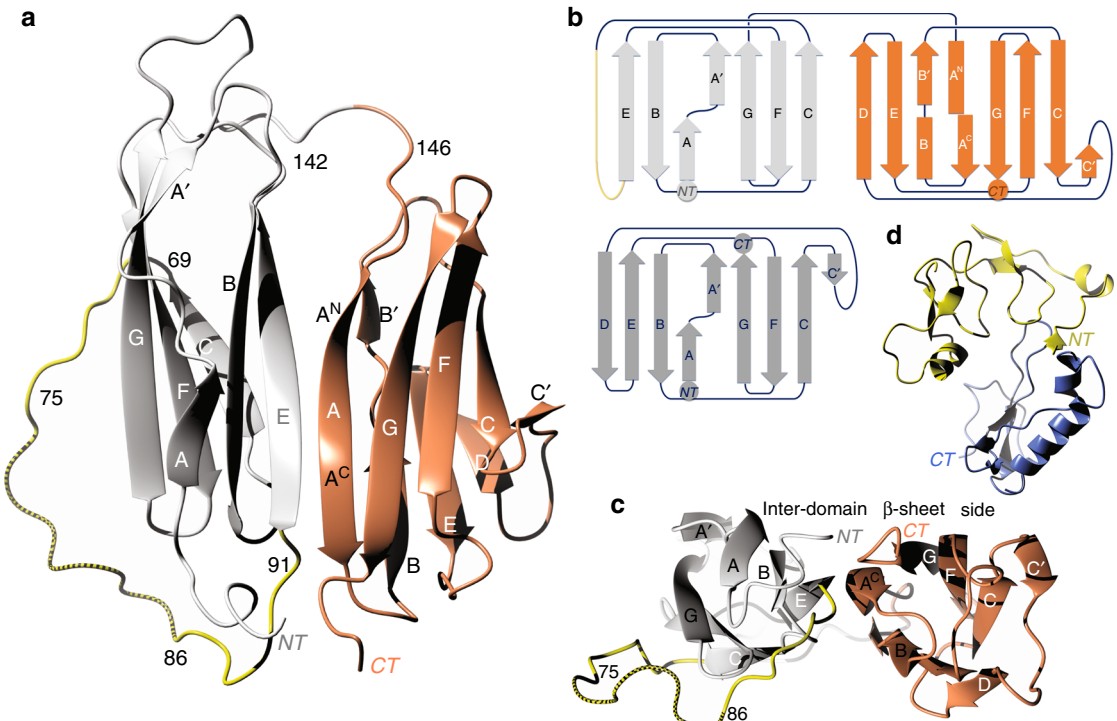

**Fig. 3** Structure and fold of the apo Imp-L2 protein. **a** Ribbon representation of the apo Imp-L2 structure. The Ig-NT and Ig-CT domains are in white and coral, respectively. NT-terminal and CT-terminal of protein. This is a top representation of the Imp-L2 with the view on a large, continuous inter-domain β-sheet. Amino acids 142–146 constitute short domains-linking loop, and 70–92 a long Ig-NT loop (in yellow); 75–86 part of this loop is marked by dashed lines, as it was highly mobile in the apo-Imp-L2, and is not included in the final model; it was traced here in a weak, non-refined electron density for an overall clarity of this figure. **b** The folding scheme of the Imp-L2 (top) and a typical Ig-I type immunoglobulin fold (below Ig-NT scheme). The grey/coral colouring scheme corresponds to the Ig-NT and Ig-CT domains, respectively. The shift of the A$^N$/A$^C$ parts of the A strand in the fold scheme of the Ig-CT domain represent a significant twist of this β-strand and split contribution of this strand to different β-sheets. **c** A view on the Imp-L2 from the N- and C-termini edge of this protein. **d** A representative structure of human IGFBP-4 (PDB ID: 2dsr); only the N-domain and C-domain are shown, and the IGF-1 has been omitted for figure clarity

follows more closely the M10 fold (pdb id: 3q4o and 2y9r, unpublished), with Cα atoms rms difference of 1.11 and 1.17 Å; the Ig-NT$^Δ$ domain has a slightly higher structural deviation of ~1.59–1.48 Å from these M10 structures.

Most importantly, Imp-L2 does not have any structural relationship to any IGFBPs (Fig. 3d) and represents a new protein scaffold for the regulation of bioavailability of the insulin-like hormones in invertebrates.

**Oligomeric states of apo-Imp-L2 in the solid state**. There are two, closely packed apo-Imp-L2 molecules in the crystal asymmetric unit, raising the question about the functional relevance of this quaternary structure. The dimer results from top-to-top-side arrangement of the Imp-L2 proteins related by twofold non-crystallographic symmetry axis that runs approximately parallel to βA:βE inter-domain interfaces (Supplementary Figure 3). The ~909 Å$^2$ buried apo-dimer interface involves mostly hydrophobic, π-cation, and only few hydrogen bonds (e.g. 37Lys −231Asp) interactions. This dimer engages more the top surfaces of the apo-Impl2 that are closer to the N- and C-termini of the id-β-sheet.

**Structure of the Imp-L2:DILP5 complex**. The Imp-L2:DILP5 complex revealed 1:1 Imp-L2:hormone mode of binding (Fig. 4). Imp-L2:DILP5 complex formation is correlated with the propensity of Imp-L2 to form a new type of the dimer. The crystal asymmetric unit contains here two independent, but practically identical dimers (rms of 1.0 Å), that are entirely different from the

apo-Imp-L2 dimer (discussed further in detail in relation to the Imp-L2 apo→holo related quaternary transitions).

The hormone:Imp-L2 binding mode is identical in all holo-Imp-L2 dimers, with the rms differences of 0.68–1.09 Å between all Imp-L2:DILP5 binary complexes. The overall folds of the holo- and apo-Imp-L2 also remain similar (~0.83 Å rms between Cα atoms).

The DILP5 hormone binds to Imp-L2 top id-β-sheet mainly by its B-helix that is almost perpendicular to the direction of id-β-strands (Fig. 4a). The helix runs in Ig-CT→Ig-NT direction, and is mostly engaged with the N- and C-termini part of the id-β surface. The hormone B-chain α−helix:Imp-L2 β-sheet interactions are depleted of specific, tight and directional side-chain:side-chain contacts, which are mostly of a van der Waals, hydrophobic nature over ca. ~835 Å$^2$ buried interface (Supplementary Table 1). The Imp-L2 contributes βB-βE:βA$^C$-βG-βF strands to the complex (mainly Trp32, Met58, Ile93, Leu159, Trp211, Met214, Phe233, Tyr215), while B-helix side chains are limited here to AlaB9, MetB13, ValB16, AlaB17. However, there is also more specific shape complementarity between DILP5 B-chain and the id-β-sheet (Fig. 4b) as, for example, hormone MetB13 fills a hydrophobic cavity formed by Met58, Trp32, Ile93 and Leu159 of the Imp-L2.

The A-chain of the DILP5 'overhangs' the edge of the id-β-sheet, contributing (e.g. by PheA16) to some van der Waals-hydrophobic interaction with Imp-L2 (Fig. 4a, b). The C-end of hormone A1 helix rests on Trp211, and the CysA9 CO–OH Tyr235 hydrogen bond (3.3 Å) is a rare directional contact on the hormone:Imp-L2 interface. However, some possible polar

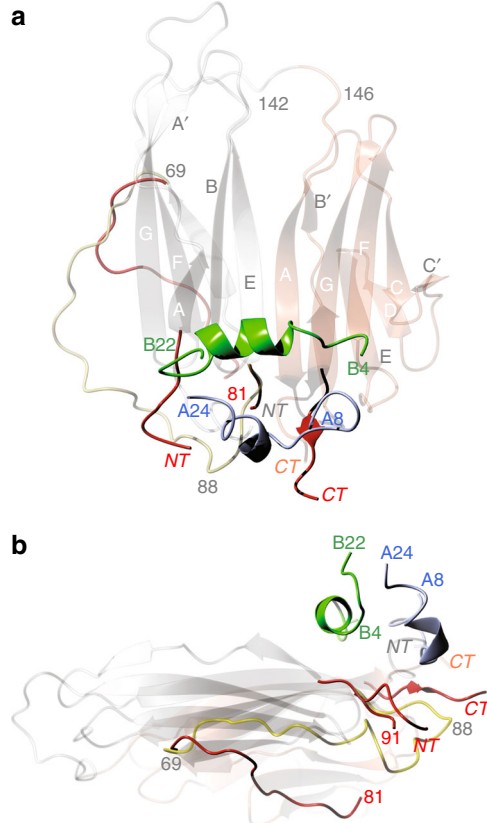

**Fig. 4** Structure of the Imp-L2:DILP5 complex. **a** The top view, and **b** the side view (from the direction of the 70–92 loop of the apo-Imp-L2) of this complex. As the core of the apo-Imp-L2 remains practically invariant upon hormone binding the main differences between apo- and holo-Imp-L2: move of the 70–92 loop and split of the C-terminal and N-terminal, have been showed in red on the apo-Imp-L2 scaffold, with a general colour-coding as in Fig. 3. DILP5 B- and A-chains are in green and blue-grey, respectively

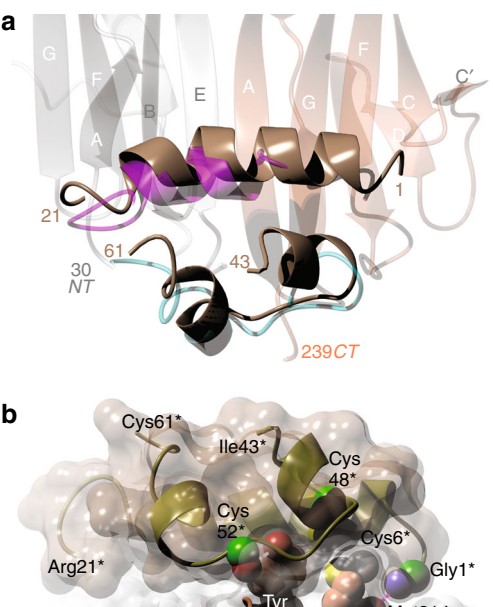

**Fig. 5** Structure of the Imp-L2:IGF-1 complex. **a** Comparison of DILP5 and human IGF-1 Imp-L2 binding modes, within the hormone binding area of the Imp-L2. IGF-1 is in brown, and DILP5 in magenta (B-chain) and light blue (A-chain); NT and CT are the apo-Imp-L2 termini and colour coded as in Fig. 3. **b** A close-up on some IGF-1:Imp-L2 major binding interaction seen from the direction of the Imp-L2 N-terminal and C-terminal. IGF-1 B 1–21 helix has also a light brown Van der Waals surface, while its A-segment is in gold only. IGF-1 amino acids numbers marked with stars. Imp-L2 Van der Waals surface in white. IGF-1 N-termini capping hydrogen bond to Imp-L2 CO group of Met214 in dash magenta line, and IGF-1 Cys52 CO hydrogen bond to OH of Imp-L2 Tyr235 in a white dash line

interactions cannot be excluded as a positively charged region of the Imp-L2 consisting of Arg95, Arg97 and Lys156, is in the proximity of AspB12, which is disordered on the apo-Imp-L2 surface (a summary of hormone:Imp-L2 interaction is provided in Supplementary Table 1).

Importantly, both the hormone and the Imp-L2 molecule undergo significant structural changes upon formation of their complex. Firstly, the conformation of the N-terminus of B-chain (B1–B5) of DLP5 must change, as its extended—so-called T-like—structure, observed in its free state[39] and in some insulins (Fig. 2a), would clash with the id-β-sheet surface. Although the terminal B1–B5 main chains of the hormone show varying degree of disorder in four copies of this complex, there is a strong indication that they attain more so-called R-like, fully helical, conformation of the B-helix, observed only in some oligomeric states of insulin, where they are induced by phenolic-like ligands[40,41] (Fig. 2a). The B19–B23 part of the DILP5 B-chain retains the conformation of the free hormone but its C-terminal part is disordered.

The fully helical transition of the B1–B5 terminus of the DILP5 B-chain upon Imp-L2 binding correlates with the reallocation (by ca. ~4.5 Å) of the hormone A-chain to avoid its clash with the Imp-L2 surface. Here, the ~3.5 Å move of CysB6 pulls the CysB6–CysA10 disulphide, and, subsequently, the whole A-chain towards its steric hindrance-free conformation.

The allosteric effect observed in the complexed hormone is reciprocated by the structural changes in the Imp-L2. The positioning of the hormone on the id-β-sheet leads to a greater separation of the N- and C-termini of the Imp-L2, from ~12 Å in apo-Imp-L2, to over 20 Å distance between the Cα atoms at sites 30 and 240 (Fig. 4a). The mutual hormone and Imp-L2 structural adaptations lead to closer contacts of the Imp-L2 236–239 C-terminal segment and the A12-A18 linker of the A-chain α-helices of DILP5.

**Structure of the human IGF-1:Imp-L2 complex.** The human IGF-1:Imp-L2 complex also reveals a dimer that is very similar to DILP5:Imp-L2 quaternary arrangement (~0.89 Å rms between Cα atoms of these complexes), and the overall IGF-1 and DILP5 Imp-L2-binding modes are very similar as well (Fig. 5a). However, a higher resolution of the X-ray data allowed here an unambiguous tracing of the IGF-1 Gly1-Cys6 N-terminus that corresponds to a partially disordered N-end of the DILP5 B-helix.

The IGF-1:Imp-L2 complex confirms the nature of the hormone:Imp-L2 interactions observed in the DILP5:Imp-L2 complex. Remarkably, the N-terminal part of the IGF-1 (Gly1-Cys6) attains clearly a previously unseen[42,43] α-helical fold, giving the B-helix of this hormone an R-state like conformation (Fig. 5a, Supplementary Fig. 4). This long B1–B19 helix is required to avoid a steric clash of this part of the hormone with the Imp-L2 id-β-sheet surface. The IGF-I R-like state is stabilised by a firm anchoring of the hormone Gly1 NH₂-terminus onto Imp-L2

surface by 1Gly-NH2—CO Met214 hydrogen bond (3.2 Å), and close Van der Waals contacts of IGF-1 Cys6–Cys48 disulphide with the side chain of Imp-L2 Met214 (Fig. 5b). Hydrogen bonds between Impl-L2 Asn216 -ND2 and -OE1 Glu9 of IGF-1 augment the conformational stability of this region. Further firm locking of the IGF-1 on the Imp-L2 surface results from the interactions similar to those observed in the DILP5:Imp-L2 complex (Supplementary Table 1). Here, the N-(1–30) and C-(Pro236-Leu238)-termini of the Imp-L2 move apart, contributing to the expansion of the id-β-sheet surface by accommodating the Asp160-Met162 peptide as a more integral part of this β-sheet. A hydrogen bond between hydroxyl of Tyr235 and peptide carbonyl of IGF-1 Cys52 is formed in this process as well (Fig. 5b). However, the end of the IGF-1 B-domain (22–29), the whole C-domain (Gly30–Gly42), and D-domain (Ala62-Ala71) residues are untraceable here, being also fully depleted of any stabilizing crystal contacts.

The R-state of hormone B-helix is correlated with other significant changes in the structure of bound IGF-1. While a typical T→R transition in human insulin leads to ~1.3 Å change in the position of CysB7 Cα atom, it doubles to 2.6 Å in IGF-1 for its corresponding Cys6. Moreover, as the CysB7-CysA7 disulphide in insulin adjusts easily to R-state α-helix, its hIGF-1 counterpart —Cys6-Cys48 cystine—isomerises in this transition, pulling Cα atom of Cys48 by ~2.7 Å, and bringing the whole Cys6–Cys48 bridge into a close contact with the Imp-L2 surface. The other IGF-1 Cys18–Cys61 disulphide also shifts by ~3.1 Å and isomerises.

The rearrangements of the disulphides in hIGF-1 that are induced upon hormone:Imp-L2 binding are propagated further by dragging the equivalents of the A-chains of the hormone—e.g. by ~7.5 Å for Leu54—from their free-hormone conformation (Supplementary Figure 4). All these changes give a curvature to the B-helix, especially around its Phe16-Arg21 region (Fig. 5a).

The importance of the Imp-L2 C-terminus (235–242) in insulin-like hormone binding is even more evident in the hIGF-1: Imp-L2 complex, in which the 51–53 linker of the hIGF-1 A-helices is hydrogen-bonded to the Imp-L2 by 52Cys CO-HN Leu238 (2.8 Å), and Ser51 OG-OC Pro236 (2.9 Å) interactions.

**Imp-L2 apo-holo quaternary transitions in crystal structures.** Although the separation of the N- and C-termini of the Imp-L2 upon binding of both DILP5 and IGF-1 is clear, and required for its effective engagement with these hormones, the largest rearrangement of the Imp-L2 tertiary structure upon apo→holo

transition is associated with the formation of its new dimer. Here, the apo-Imp-L2 top-to-top surface dimer (Supplementary Figure 3) is replaced by a much tighter back-to-back dimer (~1670 Å² buried surface), with almost perpendicular directions of the β-strands in the adjacent monomers (Fig. 6a).

The pressure of hormone A chain on the Imp-L2 C-terminus drives it away from its N-terminal end, priming in this process the Imp-L2 237–242 region for direct interactions with the hormone (Figs. 4a, 5a). Subsequently, the 1–16 part of the Imp-L2 N-terminus must move, pushing the adjacent 84–91 part of the 70–90 loop to accommodate the incoming A chains of the hormones. This, likely, triggers a large sway of the whole 60–91 loop, facilitating in this process a holo-dimerisation of the Imp-L2 that is different from its apo-dimer. Therefore, the swing of the 70–92 loop is one the key features of the apo→holo Imp-L2 transition, as this βC- and βE-strand joining protein chain that runs outside the βA-βE edge of the apo-Ig-NT domain, folds now onto the back-surface of the holo-Imp-L2 (Fig. 6b, c). However, the 70–90 loop does not stick to the back-surface of its parental Imp-L2 molecule, but is tethered into its dimeric partner molecule, running close to its βE:βB'-βC strands surface (Figs. 4a, 6b, c). This remarkable displacement of the 70–90 loop is exemplified by over 20 Å distance between the apo-/holo-Cα atoms of Asp78. The interface of the monomer-A-loop:monomer-B-bottom surface has a mixed hydrophobic/hydrophilic character, with only few hydrogen bonds, such as Arg74CO-NE2-His104 (3.4 Å), Leu80 NH-CO Val172 (2.38 Å), and Asp79 OD1-NH Val172 (3.35 Å).

**Imp-L2 self-association in solution.** The dynamics of the apo→holo Imp-L2 oligomeric transitions in solution was assessed by SEC-MALLS analysis and SAXS. In contrast to crystal-observed holo-dimerization of the Imp-L2, the addition of DILP5, insulin X14 and IGF-1 showed apo-dimer→holo-monomer phenomena in SEC-MALLS experiment (Supplementary Figure 5). As insect metamorphosis is associated with osmotic variability of the hemolymph[44,45] the quaternary behaviour of apo-Imp-L2 was also monitored by SEC-MALLS at different ionic strengths (Supplementary Figure 6). Here, the apo-Imp-L2 remained dimeric at 50 mM NaCl, formed mixed dimer/monomer populations at 150 mM NaCl, becoming prevalently monomeric at 300 mM NaCl.

The in-solution dynamic nature of the Imp-L2 was assessed further by SAXS. It confirmed that the Imp-L2 solutions were not monodisperse, with apo-Imp-L2 showing a higher apparent

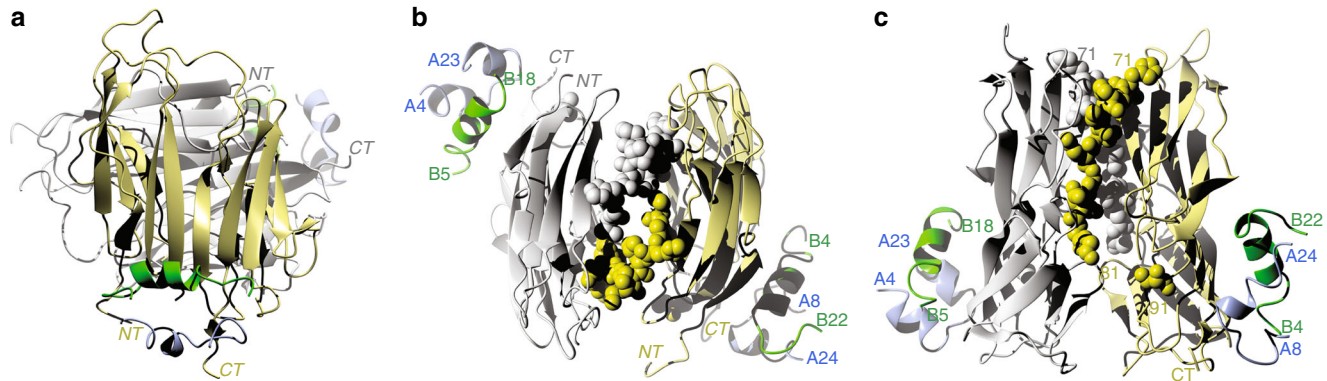

**Fig. 6** Organisation of the holo-Imp-L2 dimer in complexes with DILP5 and IGF-1. Only DILP5 complex is shown as a representative example. Each individual Imp-L2 molecule is in white and yellow, DILP5 B-chain in green and A-chain in blue. **a** The general arrangement of the holo-dimer in a view similar to as in Fig. 3a and 4a. **b** A close-up on the inter-monomer interface in holo-dimer after ~90° rotation along horizontal axis of the dimer shown in **a**. **c** A close-up on the inter-monomer interface in holo-dimer after 90° rotation along vertical axis of the dimer shown in **a**. The 70–92 loops of each dimer shown in Van der Waals spheres, and coloured correspondingly to their monomers

radius of gyration $R_g$ than its hormone complex, in agreement with the SEC-MALLS data (Supplementary Figure 7). These suggest either a change in the oligomeric state—e.g. dimer to monomer—upon the apo→holo Imp-L2 transition, or a compaction of the Imp-L2 structure upon ligand binding.

**Imp-L2 hormone binding**. Although our previous DILP5:Imp-L2-binding assay showed high-affinity interaction between these proteins[39], the precise $K_d$ of this interaction was not derived there due to a high variation of the measurements and some high non-specific binding characteristic of the used polyethylene glycol (PEG) 8000 radioactive ligands assay. Hence, the $K_d$ of the DILP5 was assessed here by isothermal titration calorimetry (ITC), and was also repeated by ITC for insulin and IGF-1, to assure some comparability of the $K_d$s of these hormones (Table 1, Supplementary Figure 8). Despite some differences resulting from the unrelated natures of PEG and ITC methodologies both types of assays showed similar ranges and trends of hormone:Imp-L2 interactions. As expected, DILP5 is the strongest binder in the ITC assay—8 nM, followed by IGF-1—13.6 nM, and insulin—135 nM.

Interestingly, thermodynamic profiles of these ITC data revealed different thermodynamic sub-processes that drive hormone:Imp-L2 binding (Supplementary Figure 9). In contrast to a prevalently enthalpic aspect of DILP5, IGF-1—Imp-L2 associations, insulin interaction with Imp-L2 shows large entropic component, which may indicate substantial, structurally challenging rearrangements on the hormone side.

To account for the variability of the ionic strength during insect metamorphosis[44,45] DILP5:Imp-L2 binding was also measured as a function of salt concentration (Supplementary Figure 10). Remarkably, the low 50 mM NaCl concentration has an inhibitory effect on hormone binding, while the high—300 mM—salt concentration shifts DILP5 $K_d$ from ~8 nM into ~5 pM range.

Finally, the ITC binding studies were complemented by SPR analysis of hormone:Imp-L2 interactions. All four hormones: human insulin and IGF-1, DILP2 and DILP5 bound to the immobilized Imp-L2, albeit with different binding kinetics (Supplementary Figure 11). However, their complex SPR characteristics did not allow fitting of the appropriate $K_d$s. Nevertheless, the SPR data confirmed the functionality of all protein components. The amount of the hormones remaining in the complexes with the immobilized Imp-L2 indicate higher DILP5 and DILP2 affinities (in a nM range) than human insulin and IGF-1, in agreement with our previous and current binding data.

## Discussion

The Imp-L2 protein represent an alternative paradigm in the molecular control of biological actions of insulin-like hormones. Both the Imp-L2 structure and its hormone binding modes show no relation to any of the human IGFBPs. Although the Ig-NT-domain and CT-domain of the Imp-L2 resembles Ig-I fold with similarity to M10 domain of human muscle protein titin, the overall arrangement of these domains, hence the overall structure of the Imp-L2, is different.

The apo→holo-transition of the Imp-L2 allows comparing its hormone binding mechanism to a molecular mouse trap. The hormone approaches the dimeric Imp-L2 trap, sensing the bait in the form of its closely located N-/C-termini (1–26, 237–242, respectively). The subsequent push of the hormone A chain-helices 51–54 linker region on the bait drives the Imp-L2 N-/C-termini apart, triggering the release of the spring of the trap in the form of 70–92 loop, which then moves onto the back side of the Imp-L2, and can stablise a new type of dimer such as observed in the crystal state.

The importance of the C-terminal region in hormone binding is corroborated by Ala-scanning of this very conserved region in Imp-L2 homologous *Sf* IBP (see Fig. 7 below), where Ala-substitutions of Phe234 (Imp-L2 Phe233), Tyr236 (Tyr235) and Pro237 (Pro236) reduce hormone affinities of the *Sf* IBP >5 fold[46].

The mechanism of Imp-L2-mediated immobilisation of the ILPs is very different from the human IGF:IGFBPs-binding mode, where a tight hormone binding is assured by the cooperation of the flexibly-linked, cleft-like N- and C-terminal domains of the IGFBPs (Fig. 3d), into which a wedge-shaped IGF is tethered via its 'edge' B-helix. In contrast, the Imp-L2 hormone binding is fulfilled by a straight accommodation of the IGF-1/DILP5 B-helix across the inter-domain β-sheet, and a large swing of the Imp-L2 70–92 loop that facilitates new dimeric quaternary arrangement of the two Impl-L2 in the crystal.

Interestingly, DILP5 affinity for the Imp-L2 is strongly ionic-strength dependent, shifting from ~8 nM at 150 mM NaCl, to ~1 pM at 300 mM NaCl. While the ITC method can be erratic at a sub-nanomolar binding range, it is, nevertheless, rather clear that DILP5:Imp-L2 affinity is significantly enhanced at high ionic strength. However, it remains unclear whether this affinity shift results from high ionic strength→monomeric Imp-L2 effect—hence higher exposition of hormone binding surface that is obstructed in the apo-dimer, or whether it reflects a physiological role of the DILP5/Imp-L2 in insects. Ecdysis, the process of shedding the outer cuticle in insects metamorphosis, is stimulated by 20-hydroxyecdysone, which induces the *Imp-L2* gene (29–30), and is paralleled by increase of the osmolarity of the moulting fluid/hemolymph[44,45]. Further investigation of Imp-L2 involvement in ecdysis is required to assess the physiological reasons behind its variable affinity to DILP5 observed in vitro.

There are discrepancies between firmly packed holo-Imp-L2 dimers that are observed in the crystal state, and the monomeric forms of the Imp-L2 complexes that dominate solutions of this protein in vitro. It cannot be excluded that the monomer↔oligomer equilibria may be shifted towards dimers upon crystallization by high Imp-L2 concentrations used during this process. Nevertheless, a tight nature of the holo-Imp-L2 dimers, a possible inter-molecular cross-bridging role of the 70–90 loop, the rearrangement of which is associated with other pronounced hormone-induced structural changes in the Imp-L2, and, finally, a higher salt crystallization environment that, according to the SEC-MALLS results, should assure a more monomeric population of Imp-L2, may still suggest some physiological relevance of its alternative dimerisation.

**Table 1 Hormone binding data for Imp-L2 and other putative IBPs from *Spodoptera frugiperda* (*Sf*-IBP) and *Trichoplusia ni* (*Tn*-IBP)**

| | *Dm* Imp-L2 | *Sf*-IBP[b] | *Tn*-IBP[b] |
|---|---|---|---|
| Insulin X14 | 135(±16)[a]/81[b] | 0.07 | 0.02 |
| Proinsulin | nm /87[b] | 0.02 | 0.01 |
| IGF-1 | 13.6(±4) [a]/17[b] | 0.17 | 45 |
| IGF-2 | nm/42[b] | 0.37 | 194 |
| DILP5 | 8.0(±1)[a]/nm | nc | nc |

*nc* not calculated to due variability of the measurements, *nm* not measured, (±) standard deviation of the ITC measurements
[a]Indicates $K_d$ values determined in this work by isothermal titration calorimetry (ITC)
[b]Indicates affinities (EC$_{50}$) measured by PEG assay in our previous work[25] (all in nM). The *Sf*-IBP and Imp-L2 proteins were obtained in ref. [25] from conditioned medium from BHK cells overexpressing these proteins, and *Tn*-IBP was from conditioned medium from *Trichoplusia ni* HI5 cells. Insulin fully monomeric X14 (ProB28Asp) mutant was used for the ITC

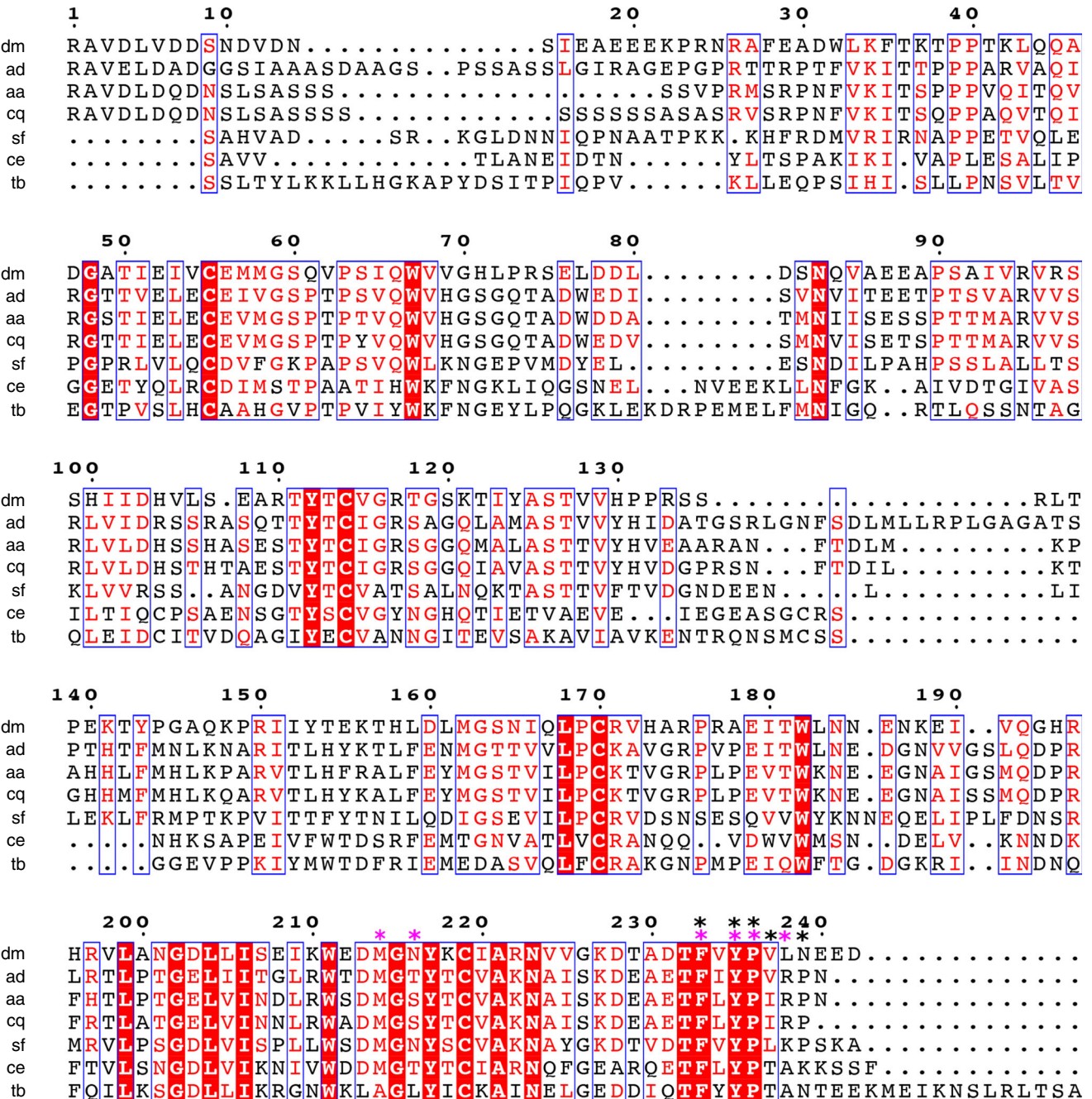

**Fig. 7** Sequence alignment of some representative and putative IBPs. Sequence numbering as for the Imp-L2 structure; key residues for hormone binding observed in the holo-Imp-L2 structures are indicated by stars in magenta, while residues that lowered *Sf*-IBP hormone affinity more than five times in Ala scanning experiment are marked by black stars. dm *D. melanogaster* Imp-L2, ad *Anopheles darlingi*, aa *Aedes aegypti*, cq *Culex quinquefasciatus*, sf *Spodoptera frugiperda*, ce *Caenorhabditis elegans*, tb *Trichinella britori*

An overall high sequence identity (33%) between *Sf* IBP and *Dm* Imp-L2 (Fig. 7[47]), especially >60% in the C-terminal ~200–240 hormone-binding regions (Supplementary Figure 12), suggests that their overall folds and hormone binding modes may be similar, and be representative of other members of insect IBP family. Therefore the differences in specificity of hormone binding (e.g. insulin vs. IGF-1) between *Sf* IBP and Imp-L2 (Table 1) may originate from more subtle side chain differences in some specific/complementary regions of IBPs and hormones, rather than from their very different tertiary structures; an ease of attainment of the R-conformation by the N-termini of the hormones can play a role here as well (see below).

The analysis of emerging genomic data, especially from databases for blood feeding insect vectors[48] indicate a presence of similar Imp-L2-like IBPs in insects. For example, Imp-L2 sequence has a high (>38%) identity with putative IBPs of *Aedes aegypti*, *Anopheles darlingi* and *Culex quinquefasciatus* mosquitos, that are responsible for spread, among others, of the dengue, zika and yellow fevers (Fig. 7, Supplementary Figure 12). Such high sequence similarity, particularly within C-terminal region of the IBPs (>40%) that is responsible for the Imp-L2: hormone interactions, suggests possible similar fold and hormone binding mode within a broader family of insects IBPs. This is of both fundamental and bio-medical significance in the light of

growing evidences about the role of ILPs in the blood-feeding-host:pathogen interactions[49], which add human hormones into a currently binary insect-vector:pathogen relationship. It seems that human insulin and IGF-1/2 present in the blood-meal ingested by an insect vector have hormone-specific effects on both the vector and pathogen physiology (e.g.[50–54]), as IGF-1, in contrast to insulin, extends the lifespan of mosquito *Anopheles stephensi*, also enhancing the resistance of this vector to *P. falciparum*[55]. Therefore, the Imp-L2:hormones complexes reported here provide the structural insight into a possible nature of some components of such complex cross-species human hormone:vector:parasite inter-relations. The lack of analogous, IBP-like, hormone control in humans makes it tempting to also probe this system to modulate vector biology in order to reduce, or block, parasite transmission.

Moreover, it would be interesting to expand the studies on IBPs on other ILPs-dependent invertebrates such as *C. elegans* and roundworm *Trichinella britori*, as they may use Imp-L2-like proteins, or similar structural modules, for their development and physiology (Fig. 7). The analysis of the available genomes suggests the extension of the previously postulated TFxYP (~242–246 (Imp-L2 numbering) signature sequence of Imp-L2 and *Sf*-IBP[46] for at least two other in the C-terminal hormone binding region of the IBPs: G-(D/E)-L-alkyl-I (~202–206), and WxDMGxYxC-(I/V)-A-(R/K)-N (~214–222) (in brackets: alternative residues in this position) (Fig. 7).

The Imp-L2 structures expand the already abundant super-family of the Ig-fold—the most coded metazoan module[56]—with further structural variations and functional application. The fusion of two Ig-domains in Imp-L2, and, possibly, in other insects IBPs, is different from similar motives found in even structurally very related human muscle protein titin M10 domain, and its complexes that rely on extensive Ig–Ig domain interactions. Their head-to-tail arrangement in M10 complexes, likely dictated by the directionality of these filaments in the muscle M-band, is different from the Imp-L2 mirror-image-like Ig-NT:Ig-CT fold.

The complexes of Imp-L2 revealed also a full B-helix R-state of the insulin-like hormones, which was considered to be limited to possible storage, or non-physiological, forms of human insulin, and it was never observed (or postulated) in IGF-1/2, nor in the known complexes of these hormones with IR. Here, we show that sole protein-protein interactions, unlike the insulin organic ligand-induced ones, are sufficient to facilitate and stabilize a full (B1–B19) helix formation in IGF-1 and DILP5. The role of phenolic-like ligands that induce the R-state in human insulin is fulfilled in the Imp-L2 by its inter-domain β-sheet surface. This demonstrates that the R-state of insulin-like hormones may not be a structural artefact, but a fold relevant for their physiology. The high entropic component observed by the ITC in the insulin:Imp-L2 complex formation correlates also with its lowest affinity of all hormones measured here. They may reflect the resistance of the required T→R conformational change of insulin B-chain upon its Imp-L2 binding due to its length, and insulin particular, B1-B6 sequence. Symptomatically, crystallizations of the insulin:Imp-L2 complex were (yet) unsuccessful.

The hIGF-1/DILP5:Imp-L2 complexes also raise a question about the origins of the modes of insulin-like hormones interactions with their receptors (i.e. IR, IGF-1R, dmIR, etc.). The structures of insulin and hIGF-1 complexes with so-called site 1 of the IR, that is located at the N-terminal part of the receptor in the form of the L1-CR domains, show mostly a non-direct hormone binding to L1 β-sheet surface (L1-β$_2$) (Fig. 8b). It is mediated there by so-called α-CT segment: an α-helical, ~16 amino-acid (704–719) C-terminus of the opposite α-subunit of the IR, and which is an integral part of site 1[57]. Although the L1

domain and Imp-L2 structures are not easily comparable, they possess similar, extensive β-sheet surfaces which both proteins use for the recruitment of an α-helix (Fig. 8a, b). In the Imp-L2 it is the long (1–19) B-helix of the hormones, while in the IR it is the α-CT segment of this receptor. As the α-helix:β-sheet inter-actions are common[58], it is interesting why IR (and IGF-1R) L1-β$_2$ surfaces are not engaged directly with B-helices of the hormones, but the α-helical CT-segments of the receptor(s) are used as mediatory components of hormone binding. Nevertheless, the similar mode of α-helix:β-sheet motif of protein:protein interaction in α-CT:IR and Imp-L2:hormone complexes is striking (Fig. 8), despite its very different functional role and origin. One of the possible reasons behind the general quaternary con-vergence but functional divergence of these structures, could be the incapability of the IR (IGF-1R) L1-β$_2$ surface to induce a fully α-helical, R-like state of insulin/IGF-1 B-chain/domain that could be required, as in Imp-L2, for an effective direct binding of the hormones. Such transition would be especially challenging for insulin, while shorter N-termini of DILP5 and IGF-1 are more capable of such structural change (Supplementary Figure 9). Therefore, the receptor α-CT helical segment must replace the hormone B-helix and serve as an anchor for the attachment of hormones to IR and IGF-1R. However, the different modes of hormone:protein interactions in IBPs and IR/IGF-1R may arise from many other, more function-related reasons, as, for example, a non-direct hormone binding to the L1-β$_2$ surfaces of receptors, facilitates modulation/attenuation of the hormone binding, hence signal transduction, helping also to maintain the receptors in on/off states. A complex signalling of the IR-like receptors must reflect the multifaceted regulatory roles of insulin and IGF-1/IGF-2, and a direct hormone binding to the receptor L1-β$_2$ sur-face would be too simplistic for this purpose.

In summary, the relevance of structural and functional insights into apo/holo-Imp-L2: IBP system reported is manyfold. Firstly, it reveals a β-sheet fold that serves as a direct, high-affinity binder of these hormones. Secondly, Imp-L2 represents likely an alternative hormone binding and regulatory IBP system that is different and evolutionarily independent from human IGFBPs. Moreover, Imp-L2 shows capability of enforcing an allosteric effect on insulin-like hormones, inducing R-state conformation of their corresponding B-chain α-helices, a phenomenon not observed before in any insulin-like hormones:protein interactions. A similar α-helix:β-sheet mode of hormone:protein interface that is employed by both Imp-L2 and IR/IGF-1R, sheds also light on a possible molecular reasons behind the functional divergence of these two systems. Finally, a likely structural conservation of Imp-L2 hor-mone-binding mode in other IBPs in insects and in some other invertebrates, together with their possible interactions with human insulin and IGF-1/2 in insect vectors, expands the rele-vance of these IBPs towards insect-vector associated diseases, indicating also opportunities for the exploration of other approaches in pathogen transmission-blocking strategies.

## Methods

**Production of recombinant proteins**. The construct encoding Imp-L2 recombi-nant protein was made using polymerase chain reaction amplification of *Droso-phila* Imp-L2 cDNA (Uniprot No: Q09024) available in-house[25] for sub-cloning into the pBac4x vector (Novagen) using the SLIC method[59]. Briefly, the insert and backbone fragments were amplified with 15 base pair complementarity on the 3′ and 5′ ends of each fragment. Baculovirus was produced using the FlashBac method (Oxford Expression Technologies) and bacmid[60] and purified from the conditioned media of infected Sf9 cells (primers are listed in Supplementary Table 2). The recombinant protein consisted of the full-length protein with the native signal peptide and C-terminal HRV14 3 C proteolytic site (EVLFQGP), followed by two SR residues and a Strep-tag II (SAWSHPQFEK) sequence. In the first step of the purification the buffer was changed to PBS (Invitrogen) by gel-filtration of 1 L supernatant using an INdEX 100/95 column (GE Healthcare). The resulting sample was run on a Strep-Tactin column (IBA), and the protein was

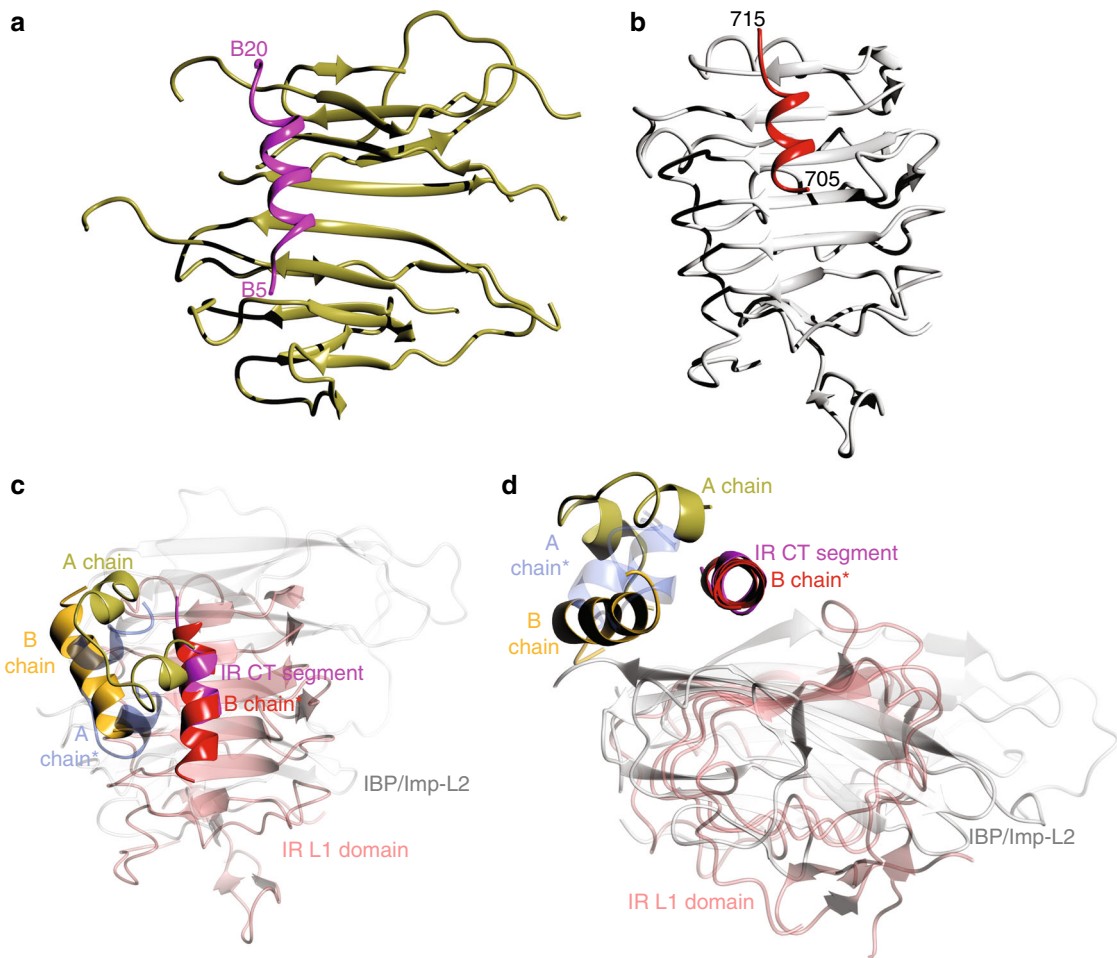

**Fig. 8** Comparison of DILP5:Imp-L2 and insulin:IR binding modes. **a**, **b** Are the top views on the Imp-L2 and L1 IR surfaces, with shown only the directly bound B-helix of DIPL5 (in magenta), and the IR CT-segment (in red), respectively. **c**, **d** Superpositions of the Imp-L2:DILP5 and L1 IR:CT:insulin complexes (PDB ID 3w12), IR L1 in pink, Imp-L2 in white, IR CT-segment in magenta; DILP5 B- and A-chains are marked by red and blue stars, respectively; human insulin in IR complex coloured in yellow (B-chain) and green (A-chain). The complexes in **d** are more in a close-up after ~90° rotation of the view in **c** along its horizontal axis

eluted by 2.5 mM desthiobiotin (Sigma-Aldrich) in PBS, with the yield ~ 5 mg/L of cell culture. The Strep-tag II was removed by available in-house HRV14 3C protease[61]. Finally, the gel-filtration with HiLoad 16/60 Superdex 75 prep grade column (GE Healthcare) were performed to change either into 10 mM HEPES (Sigma-Aldrich) pH 7.4, 20 mM NaCl (Sigma-Aldrich) crystallization buffer, or into PBS buffer for further binding/SAXS studies. The selenomethionine-labelled protein was produced according to Bellizzi et al.[62], with the yield ~ 1 mg/L of cell culture. The complex of Imp-L2 with DILP5 for SAXS experiments was purified by running 1 mL mixture of 5 mg of Imp-L2 and 15 mg of DILP5 available in-house[39] on a HiLoad 16/60 Superdex 75 prep grade column. N-terminus one amino acid truncated (B2-B29), expression optimized, so-called C4 variant of DILP5[39] has been used throughout this study as it is referred here as DILP5. Human IGF-1 used for co-crystallization and binding experiments was available in-house (Novo Nordisk). Synthetic DILP2 for SPR binding experiments was prepared as previously described[63].

**Hormone-Imp-L2 binding ITC and SEC-MALS studies**. Experiments were carried out using a MicroCal200 calorimeter (Malvern). Imp-L2 (0.5–0.7 mg/mL) was in 50 mM Tris, 150 mM NaCl, pH 7.4 buffer. Imp-L2 concentration was measured by UV using an extinction coefficient of $E_{280} = 1.25$ cm$^2$/mg. Typically, a run consisted of 19 injections of 2 µL into the cell with a 30 s interval between injections. Duplicates of all experiments were carried out at 25 °C. Imp-L2 concentrations of 12.8–22.6 µM were used with a hormone concentration about tenfold greater. The concentration of Imp-L2 was adjusted to give c values (c = $K_a$[Imp-L2], where $K_a$ is the equilibrium binding constant) between 10 and 500. Very high values of c (>1000) in the case of DILP5 led to a step-shaped isotherm limiting the accuracy of the fitted $K_a$ value. Data fitting and analysis were carried out with the manufacturer's software, MicroCal PEAQ-ITC Analysis. The concentration of the

hormones were adjusted so that the stoichiometry of binding was 1:1 given that the Imp-L2 has a single binding site.

The effects of different ionic strengths on apo-Imp-L2 homodimerization were investigated by SEC-MALLS as well. In these experiments ligand free 100 µL Impl-L2 samples at 1 mg/mL in 50 mM Tris, pH 7.4 buffer were incubated in at 50, 150, and 300 mM NaCl and run on a Superdex S200 10/300 gel filtration column.

The change of the apo- to holo-aggregation of the Imp-L2 upon binding of DILP5, insulin and IGF-1 was also measured at 150 mM NaCl, 50 mM Tris, pH 7.4, by SEC-MALLS Superdex S200 10/300 gel filtration column at room temperature.

**SPR experiments**. Approximately 500 RU of Imp-L2 was immobilized on CM5 chip using amino coupling kit (Biacore/GE Healthcare). The flow speed was set to 50 µl/min. The experiments were performed in PBS buffer (Invitrogen) at 25 C using T200 Biacore instrument.

**Crystallization, crystal structure and SAXS experiments**. Crystallization of apo-Impl-L2 resulted from hanging-drop experiments by mixing 1 µL protein (5.5 mg/mL) and 1 µL reservoir solution (17% w/v PEG 6 K, 0.1 M Tris/HCl, pH 7.0). The structure was solved by multiwavelength anomalous dispersion using SHELX software[64]. Crystallization of the DILP5:Imp-L2 complex was obtained by mixing equivalent volumes of protein (10 mg/mL, at IMP-L2:DILP5 1:3 molar ratio) and reservoir solution (8–10% w/v PEG 4 K or 6 K, 20 mM MgCl$_2$, 0.1 M HEPES pH 6.8–7.5). Crystals of the IGF-1:Imp-L2 complex grew by mixing equivalent volumes of protein complex (10 mg/mL, Imp-L2:IGFI-1 1:3 molar ratio) and reservoir solution (4–8% w/v PEG 6 K, 5–20 mM MgCl$_2$, 5 mM SB12, 0.1 M Tris pH 7.5). All crystallization experiments were performed at 293 K. All crystals were directly flash-cooled in liquid N$_2$, X-ray data were collected at 100 K and processed by xia2[65]. The structures of the

complexes were solved by Molecular Replacement[66,67], using the structures of apo-Imp-L2, DILP5 (C4 version of DILP5[39]), and IGF-1 (PDB ID 1gzr) as models. Model building and refinement were performed by COOT[68] and the CCP4 suite of programs[69]. Details of X-ray data collection, and refinement statistics are in Supplementary Table 3, and representative stero images of portions of electron density maps are shown in Supplementary Figure 13. Figures were made using CCP4mg[70].

The SAXS experiments were performed at ESRF Grenoble, beamline ID14-3. The samples were in PBS buffer (Invitrogen) and SAXS data were recorded at 25 °C during a $10 \times 10$ s exposure time on a 1 M pixel 2D Pilatus detector (DECTRIS) covering a $q$ range pf $0.0055 - 0.609$ Å$^{-1}$ ($q = 4\pi\sin\theta/\lambda$, where $\lambda$ is the wavelength and $\theta$ is half the scattering angle). Initial data processing was conducted using programs from the ATSAS package[71].

## Data availability

The structures of Imp-L2 and its DILP5 and IGF-1 complexes were deposited in PDB database under accession codes 4CBP, 6FEY, 6FF3, respectively. SAXS curves and experimental parameters were deposited with the SASDB (www.sasdb.org) under accession numbers SASDDS8, and SASDDT8. All other data supporting the findings of this study are available from the corresponding author on reasonable request.

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

## Acknowledgements

A.M.B., T.R.G., C.M.V. work was supported by the Medical Research Council (Grant MR/K000179/1 and MR/R009066/1 to A.M.B.). J.D.W. is supported by an NHMRC Principal Research Fellowship (1117483). We thank Maxlab (Lund, Sweden), ESRF Grenoble and Diamond Light Source (proposal numbers mx-1221 and mx-7864) for access to their beam lines that contributed to the results presented here. We thank Mr. Sam Hart for assistance with data collection, and Chris Watson for crystallizations.

## Author contributions

A.S.A., C.K., W.S. and N.K.R. have cloned and produced DILP5, Imp-L2. N.K.R., O.K., G.S., C.M.V., T.G., J.T. and A.M.B. have solved crystal structures of Imp-L2 alone and in complex with DILP5/IGF-1. J.W. produced the DILP2. G.S. and M.N. have performed SAXS analyses of Imp-L2. N.K.R., P.D.M. and A.M.B. wrote the manuscript. All authors read and approved the final manuscript. N.K.R. and C.M.V. contributed equally to this work.

## Additional information

**Competing interests:** The authors declare no competing interests.

