## [Peer Review File · Nature Communications]

Reviewers' comments:

Reviewer #1 (Remarks to the Author):

The authors describe the molecular structure of a *Drosophila* protein Imp-L2 that has previously been shown to bind not only *Drosophila* insulin-like peptides (DILPs) but also human insulin and IGFs. It was already predicted from sequence analysis that the protein possessed two domains with Ig-like folds, and that Imp-L2 was unrelated to the well-characterised family of mammalian IGF-binding proteins, and this is confirmed by the structural analysis derived by X-ray crystallography. Importantly, structures are presented not only for free Imp-L2 but also for complexes with DILP-5 and hIGF-1, allowing dissection of the conformational changes in both Imp-L2 and its ligands consequent on their interaction.

The structural data are original and are well described and discussed in detail. However the pathophysiological implications that would make the study of wide general interest receive less consideration than the title given to the paper requires.

1. The impact of the study would be enhanced if it included more data on Imp-L2/IBP ligand binding. Previously published binding data for insulin and IGFs are reproduced in Suppl Table 1 (which incorrectly refers to ref 27 – it should be ref 25), but this analysis did not include DILPs. The SPR analysis of ligand binding that is presented (Suppl Fig 4), while confirming interaction of Imp-L2 with hIGF-1 and insulin as well as DILPs 2 and 5, did not allow determination of KDs (p.11, para 2) but was suggestive of higher affinities for DILPs than insulin/IGF. It would be very interesting to see the affinities of Imp-L2 and Sf-IBP for DILPs, determined as in ref 25. There seems no reason why such data could not be generated, at least for DILPs 2 and 5. There are enormous (>1000-fold) differences in affinity of IBPs of different insect species for mammalian insulin and IGFs, and in terms of understanding the physiological role of IBPs in insects, and their potential as therapeutic targets, it is important to know whether there is more uniformity in their affinity for DILPs.

2. Data on binding affinity are especially important in relation to the discussion regarding potential interaction of host hormones with insect IBPs (p.13, para.4 – p.14, para.1). Concentrations of DILPs in insects are presumably unknown but mammalian serum concentrations of insulin and free IGFs are much lower than the quoted affinities of Imp-L2 (ref 25). On the other hand Sf IBP has a ~1000-fold higher affinity for insulin. There is a brief discussion of relative specificities of different IBPs for insulin vs IGF in terms of structural considerations (p.13 para) but no comment on the pathophysiological implications of these affinity differences.

3. If the Tn-IBP sequence is available it should be included in Fig 6. Would it be meaningful to map ligand contact residues onto the primary sequence of Imp-L2? And if so might this shed any light on the marked differences in ligand affinity for different IBPs and/or predict the likely binding properties of Aa-IBP?

Signed: Kenneth Siddle

Reviewer #2 (Remarks to the Author):

Insect insulin-like polypeptides binding proteins (IBPs) have been considered to be structural and functional homologues of the IGF binding proteins (IGFBP). This manuscript describes the first insect IBP structure, that of the *Drosophila* Imp-L2 protein, both free and in complex with *Drosophila* insulin-like peptide 5 (DILP5) and human IGF-1. These structures identify a new, Ig-fold architecture that is quite distinct from the common structures of the IGFBPs, and which binds ILPs in a different fashion.

The authors also suggest similar hormone binding modes in insect vectors such as mosquito, raising the possibility that these interactions may present opportunities to modulate the transmission or progression of mosquito-borne diseases such as malaria, dengue and yellow fever.

The Introduction is rather unfocussed. Similarly, the Discussion is too long.

p. 8: Role of electrostatic interactions in Imp-L2 - DILP5 binding: has the effect of ionic strength on binding affinity been assessed?

p. 11: in view of the failure of SPR to provide reliable K_d values, ITC could be utilised. ITC measurements would also clarify the stoichiometry.

Figure 4: I found the comparison between the DILP5 and human IGF-1 Imp-L2 binding modes difficult to follow from this figure.

Has the mechanism of release of hormone from the complex with Imp-L2 been considered?

There are numerous minor grammatical errors in the manuscript, of which the following are just a few examples:

p. 3: and some possibly some non-signalling

p. 4: with suggested ≤ 100 -fold affinity than classical IGF-BPs,

There are also several rather convoluted sentences, which make the narrative unnecessarily difficult to follow, for example:

p. 11: Moreover, the 1-16 N-terminus of the Imp-L2 must also reallocate to accommodate the incoming A chains of the hormones, thrusting the adjacent 84-91 segment of the 70-90 Imp-L2 loop. This, likely, triggers a large sway of the whole 60-91, loop-including protein chain, initiating a tight holo-dimerisation of the Imp-L2, in an arrangement that is different from its apo-dimer.

p. 15: In summary, the structural and functional insights into apo/holo-Imp-L2 IBP system reported here transgress their importance for the insect-related regulation of the bioavailability of insulin-like hormones.

p. 17: the buffer conditions and temperature for SPR and SAXS experiments should be specified.

Bibliography: all species names to be italicised.

Reviewer #3 (Remarks to the Author):

This work is very interesting and brings valuable and novel information on the structure of an insect insulin-like peptide binding protein (IBP). This study is the first to elucidate this structure in insects, thereby revealing significant differences with human IGF-binding proteins (IGFBPs). Here are some minor and major concerns:

Minor comments:

Introduction:

(1.) The authors have a strange way of showing the abbreviations. For instance: IGF-Binding Proteins (IGFBP); why are I, B and P written in italics? Perhaps it is to show where the abbreviation comes from, but this is quite obvious. Especially in this case, where also G and F are part of the abbreviation. However, the authors are not consequent in this, because a bit further (page 2) they do not only write the first character in italics, but also underline it. So, above all, try to be more consistent.

(2.) "This involves freely circulating IGFBPs that are referred here as ILPs Binding Proteins (IBPs)". It is not clear what you mean with 'freely circulating'. Furthermore, it is confusing to use another term (IBPs) when you are talking about the IGFBPs. Just respect existing nomenclature. So use IGFBPs when you are talking about human IGF binding proteins, and use IBP when you are talking about insulin-like peptide binding proteins (in organisms like insects).

(3.) "...secrete a binding protein - named Sf-IBP - that binds insulin, IGF-1 and IGF-2". I suppose you mean human (or mammalian) insulin, IGF-1 and IGF-2? Please specify this.

(4.) "Here, Andersen et al. (25) showed that Sf-IBP has high insulin and significant IGF-1/IGF-2 affinities (70, 170 and 370 pM, respectively (Table S1)), " this sentence is not correct/clear; what do you mean here with "significant"?

(5.) "...cloning indicating the presence of two Ig-like C2 domains...". Explain a bit more about Ig-

like domains. This is not immediately clear for the broader public of this journal.

(6.) last alinea: Write 3-D as 3D

Results:

(7.) "... consists of two b-sheets: bA'- bG-bF-bC and ..." ◊ remove the space between '-' and 'bG'.

(8.) "... which joins bE strand from a different/opposite..." add '-' between bE and strand (to be consistent).

(9.) "The continuous inter-domain-b-sheet (id-b-sheet, 'top-side' of the Imp-L2) is formed by a tight interface between Ig-NT bA-bB-bE and Ig-CT bAC-bG-bF-bC-bC' b-sheets that is stabilised by hydrogen bonds between parts of bE (92-99) and bA (152-160) strands." Do you mean bAC (152-160) with 'bA (152-160)'? It is confusing since the Ig-NT also has a bA sheet...

(10.) Relation of the Imp-L2 to the Ig-fold containing proteins. 'PDB' is an abbreviation used for the first time in the text. Please write in full.

(11.) "These mutual hormone and IBP structural rearrangements lead to closer contacts of the Imp-L2 236-239 C-terminal segment and A-chain A12-A18 linker of the A-chain α -helices of DILP-5." Use Imp-L2 instead of IBP to be consistent and less confusing.

Discussion

(12.) "A high sequence identity (33%) between Sf IBP and Dm Imp-L2 (Fig. 6, (45)), especially in hormone-binding regions, ...". What is the similarity of the hormone-binding regions?

(13.) "... dengue, zika and yellow fevers (Fig. 6,)." Remove comma behind Fig. 6.

Methods

(14.) "The construct encoding Imp-L2 recombinant protein was made using polymerase chain reaction amplification of human Imp-L2 cDNA (Uniprot No: Q09024) ..." Don't you mean Drosophila instead of human?

(15.) Where did you get DILP5, DILP2, insulin and IGF-1? Mention this in the methods section.

Figures:

(16.) Figure 1: You always speak about DILPs, but these are Drosophila ILPs. Make this scheme more general towards other insects as well. Or speak about Drosophila instead of insects.

(17.) Figure 2: "The Ig-NT and Ig-CT domains are in white and coral, respectively. NT and CT – termini of protein." Better to write: "The Ig-NT and Ig-CT domains are in white and coral, respectively (NT and CT – termini of protein)."

(18.) Figure 2: "The white/coral colouring scheme corresponds to the Ig-NT and Ig-CT domains." It is better to write "The white and coral colouring scheme corresponds to the Ig-NT and Ig-CT domains in a."

(19.) Figure 4: the violet colour of DILP5 is not that visible. Please use another colour.

Major Comments:

(A.) Information about the homology of DILP5 and DILP2 is missing. Are they structurally different? Is there a difference in how they bind the Imp-L2 (crystal structure)?

(B.) How similar is DILP5 to other ILPs in other insects? And thus, are these relevant DILPs to make hypotheses about ILP:IBP interactions in other insects? Explain this in the discussion.

(C.) How similar is DILP5 to IGF-1? Include this in supplementary data and discuss in text.

(D.) Why can Imp-L2 not bind other DILPs? Discuss this in your discussion and add a figure in supplementary data to show the differences/similarities between the different DILPs.

(E.) You performed SPR analysis on Imp-L2:hormone interactions. Here you tested 4 hormones:

insulin, IGF-1, DILP2 and DILP5. However, you only show crystal structures for IGF-1 and DILP5. How similar are these 4 hormones? And do you know if they have similar properties (structural) when binding Imp-L2? Wouldn't it be interesting to also investigate the crystal structure of insulin:Imp-L2?

(F.) IBPs of more 'primitive' (ametabolous and hemimetabolous) insects are missing in fig. 6. Include these also in your discussion.

(G.) Results: "As a result, the N-terminal part of this bA strand (147-154: referred here as to bAN) belongs to bANbB/bB'-bE-bD b-sheet of the Ig-CT, while the C-terminal part of this strand (157-160: referred here as to bAC) contributes to the bAC-bG-bF-bC-bC' b-sheet of this domain." It is confusing to have A in fig. 2a and Ac and AN in fig. 2b.

(H.) Results: "The relative flatness of the id-b-sheet surface contrasts with a shallow, concave shape of the opposite ('back') side of the Imp-L2, with its apex at the bA: bE inter-domain interface. There is an almost two b-strand width gap between the nearest Ig-NT and Ig-CT b-strands on the back side of the Imp-L2 which result in only one b-sheet thickness in the central part of this protein." This is not clear, clarify this in your figure.

(I.) Results: "Oligomeric states of apo-imp-l2 in the solid state." Even after reading the discussion I'm still wondering how functionally relevant this is. Can you find these oligomers in the fruit fly itself? Perform a western blot on hemolymph or on cells overexpressing this protein to see if dimers/oligomers occur in vivo.

(J.) Discussion: you talk about 'Anopheles aegypti'. But as far as I know this mosquito does not exist. Do you mean *Aedes aegypti* or *Anopheles gambiae*? *Anopheles* is also written with one 'n' and not two.

(K.) Why did you use Imp-L2 and not an IBP that has a higher affinity for insulin like the SfIBP? Or why didn't you perform this study on a more relevant insect species, like *Aedes aegypti*?

(L.) "A high sequence identity (33%) between Sf IBP and Dm Imp-L2 (Fig. 6, (45)), especially in hormone-binding regions, suggest that their overall folds and hormone binding modes are very similar, hence may be representative of other members of insect IBP family. The differences in specificity of hormone binding (e.g. insulin vs IGF-1) between Sf IBP and Imp-L2 (Table 1) would then result from subtle side chain differences in some specific regions of these IBPs, rather than from their different tertiary structures." I think this hypothesis is too strongly formulated. Tone down. Because you don't know this for sure based on the data you present in this paper.

(M.) "The Imp-L2 structures expand the already abundant superfamily of the Ig-fold - the most coded metazoan module (54) - onto their new structural variations and functional application. The fusion of two Ig-domains in Imp-L2, and, likely, in the other insects IBPs, is different from similar motives found in even structurally very related human muscle protein titin M10 domain, and its complexes that rely on extensive Ig-Ig domain interactions. Their 'head-to-tail' arrangement in M10 complexes, which may be dictated by the directionality of these filaments in the muscle M-band, is very different from the Imp-L2 'mirror-image'-like Ig-NT:Ig-CT fold." Similar as previous comment. Tone down.

(N.) "Moreover, insect IBPs show capability of enforcing an allosteric effect on insulin-like hormones, inducing ..." 'Speaking about insect IBPs is far too general. Tone this down. It may not be the case for all insect IBPs.

REVIEWER #1

The authors describe the molecular structure of a *Drosophila* protein Imp-L2 that has previously been shown to bind not only *Drosophila* insulin-like peptides (DILPs) but also human insulin and IGFs. It was already predicted from sequence analysis that the protein possessed two domains with Ig-like folds, and that Imp-L2 was unrelated to the well-characterised family of mammalian IGF-binding proteins, and this is confirmed by the structural analysis derived by X-ray crystallography. Importantly, structures are presented not only for free Imp-L2 but also for complexes with DILP-5 and hIGF-1, allowing dissection of the conformational changes in both Imp-L2 and its ligands consequent on their interaction.

The structural data are original and are well described and discussed in detail. However the patho-physiological implications that would make the study of wide general interest receive less consideration than the title given to the paper requires.

1. REVIEWER: *1. The impact of the study would be enhanced if it included more data on Imp-L2/IBP ligand binding. Previously published binding data for insulin and IGFs are reproduced in Suppl Table 1 (which incorrectly refers to ref 27 - it should be ref 25), but this analysis did not include DILPs. The SPR analysis of ligand binding that is presented (Suppl Fig 4), while confirming interaction of Imp-L2 with hIGF-1 and insulin as well as DILPs 2 and 5, did not allow determination of K_Ds (p.11, para 2) but was suggestive of higher affinities for DILPs than insulin/IGF. It would be very interesting to see the affinities of Imp-L2 and Sf-IBP for DILPs, determined as in ref 25. There seems no reason why such data could not be generated, at least for DILPs 2 and 5.*

Response: We addressed this very valid point. The DILP5 *K_d* to Imp-L2 has been measured by the ITC (new Table 1, Sup. Fig 8 and 9). As a very different PEG assay using radioactive ligands was employed before (ref 25), binding of insulin and IGF-1 to Imp-L2 was remeasured by the ITC as well (new Table 1, Sup. Fig 8 and 9) to remove the methodological bias in comparison of DILP5 ITC-derived *K_d* to the *K_d*s of other hormones measured by the PEG assay. The thermodynamic components of these binding data have been assessed as well (new Suppl. Fig 9). As expected DILP5 is the strongest Imp-L2 binder (8 nM), followed by IGF-1 (13.6 nM), and human insulin (135 nM).

Moreover, DILP5 - Imp-L2 binding was also measured as a function of the ionic strength (50, 150, 300 mM NaCl) that may roughly mimic the variations of the osmotic pressure of the hemolymph during insect metamorphosis. Remarkably, the lowest ionic strength abolished DILP5-Imp-L2 binding, while the highest ionic strength shifted DILP5 affinity from ~8 nM into a ~5 pM range (new Suppl. Fig.10).

These findings correlate further with our new SEC-MALLS data on apo-Imp-L2 behaviour in similar NaCl range, as the lowest ionic strength stabilised the apo-Imp-L2 dimer that shelter the hormone binding sites, while the highest ionic strength shifts the quaternary equilibria towards free monomer with freed hormone binding surface (new Suppl. Fig.6).

These correlations are now even further paralleled by the SEC-MALLS tracing of the Imp-L2 behaviour upon binding of the hormones which, again, clearly shift the apo-dimer into holo-monomers (new Suppl. Fig 5).

Unfortunately, we could not perform similar experiments for DILP2 as we simply ran out of this hormone. This is not a simple technical issue, as the

production of DILP-like hormones still represents formidable challenge to chemistry and biochemistry. We had a limited amount of DILP2 that was consumed by previous experiments that led to this manuscript, and we were not able to undertake an additional full chemical synthesis of DILP2 for this work. However, we are building research towards future grant applications for more in-depth studies of Imp-L2 and related IBPs, and DILP hormones production is one of the key parts of this process. Hence the issue of DILP2 availability – and of related hormones – will be fully addressed by us in the future.

2. REVIEWER: *There are enormous (>1000-fold) differences in affinity of IBPs of different insect species for mammalian insulin and IGFs, and in terms of understanding the physiological role of IBPs in insects, and their potential as therapeutic targets, it is important to know whether there is more uniformity in their affinity for DILPs.*

Response: We fully agree that the understanding of the molecular basis of enormous differences among hormones:IBPs affinities is one of the most intriguing aspects of this research, and we will address it fully in the next stage of our work on IBPs. As said above, we are in the process of gathering critical new data for a grant application that will also tackle affinities-dictating issues. We will attempt to clone and produce the relevant IBPs (*Sf*-IBPs and IBPs from different insect vectors), synthesise the required hormones and characterise structural and functional signatures of these systems. These aspects of hormone:IBP interactions are at the centre of our interest, however their proper elucidation is beyond the scope of this manuscript which is the first structure-function reporting of hormone:IBP interplay. It opens/signals many new possible venues to pursue, but it cannot answer all these questions due to technical, time, funding and manpower challenges. It is already saturated with novel interdisciplinary data.

3. REVIEWER: *2. Data on binding affinity are especially important in relation to the discussion regarding potential interaction of host hormones with insect IBPs (p.13, para.4 – p.14, para.1). Concentrations of DILPs in insects are presumably unknown but mammalian serum concentrations of insulin and free IGFs are much lower than the quoted affinities of Imp-L2 (ref 25). On the other hand Sf IBP has a ~1000-fold higher affinity for insulin. There is a brief discussion of relative specificities of different IBPs for insulin vs IGF in terms of structural considerations (p.13 para) but no comment on the pathophysiological implications of these affinity differences.*

Response: We fully agree that the pathophysiological implications of our findings and emerging evidences about interplay of human host:insect vector insulin-like hormone systems are fascinating, and a most inviting subject for a discussion. However, the research about human insulin/IGF-1/2 impact on vector physiology (and vice versa) is still at a very early developing stage, and we must fully characterise mosquito and other insect IBP systems (in plans) prior to formulation of more advanced, and research-sound, hypotheses. We have provided some initial, most interesting elements of these puzzles but must restrain ourselves in further hypotheses. In contrast, the other referees asked us to tone down the discussion on this subject. Hence, although we would like to follow referee's demands, here we will remain more modest in our opinion of the current state of the relevant science.

4. REVIEWER: *3. If the Tn-IBP sequence is available it should be included in Fig 6.*

Response: The genome of *Trichoplusia ni* has just been released but it does not yet provide unambiguous results in BLAST-like searches in the context. A protein or DNA query of dmIMP-L2 or sfIBP sequences against the *Trichoplusia ni* genome using BLASTp, PSI-BLAST and PHI-BLAST was not very successful in identifying any clear *T. ni* homologues. We tried to incorporate 'the best' *Tn*-IBP into main Figure 7 but this only diluted its current clarity. However, we provide here one of the best alignments (at the end of these responses) to satisfy referee's request. We also include a sequence from *Plasmodium vivax* that may be of some interest in this context.

5. REVIEWER: *Would it be meaningful to map ligand contact residues onto the primary sequence of Imp-L2? And if so might this shed any light on the marked differences in ligand affinity for different IBPs and/or predict the likely binding properties of Aa-IBP?*

Response: This has been done in Figure 7. However, it is difficult to predict the hormone binding properties of the vector (e.g. Aa, Ad, Cq mosquitos) as the regions of the IBPs that are critical to binding of the hormones are very conserved among these insects. Actually, we postulate that they are sequence signatures of the IBPs. We also provide a summary of the main hormones:Imp-L2 interactions in Suppl. Table 1, which underline only a few more firm, HB-mediated contacts, while the majority of the binding is facilitated of many weak van der Waals interaction of, mostly, hydrophobic nature. Hence these findings make any prediction for hormone preference quite difficult and too speculative at this stage. We will address these issues systematically in the future research programme on this subject.

```

dm .....
pv WELYKKFERNLLI.....YDKSKNFHNEWEMNEEK.....
tn .PVLKDKQAEVLFRADNYSTAPLECALEGHGKDVKYTWYKNGQIFDWEKAGHIAQRPGEG
sf .....
ad .....
aa .....
cq .....
ce .....MFAIAL.....
tb .....SS.....

```

```

          1      10      20      30      40
dm ...RAVDLVDDSDNDVDN.....SIEAEEKPRNRAFADWLFKFTKTPPTKQQ
pv .LCFYLYKW...IYDQL.....ISKKVTKTEYANFNLWDMRKSEICP
tn SIMFFNPQPSDEGQYKCL.....VETPAGIATRITLKKAFINIPKVTLQEHHRP
sf .....SAHVAD.....SRKGLDNNIQPNAATPKK.KHFRDMVRI RNAPPETVQL
ad ...RAVELDADGGSIAAASDAAGSPSSASSLGRAGEPGPRTRPTFVKITTPPPARVAQ
aa ...RAVDLDQDNLSSLSASSS.....SSVPRMSRPNFVKITSPPPVQITQ
cq ...RAVDLDQDNLSSLSASSS.....SSSSSSASASRVRPNFVKITSQPPAVTQ
ce .LSFLVVLINAHPPMHA.EMHSAVVTLANEIDTN.....YLTSAPAKIKI.VAPLESALI
tb .LTYLKKLLHGKAPYDS.ITP.....IQPV.....KLLSQPSIHI.SLLPNSVLT

```

```

          50
dm ADGATIEIVCE.....
pv TCMCEFKIKNLSNLIQLKKAYDY..YLFDAY.KKTSKINDQISD.....KNYCKYI
tn IEGKPFKLECK.....IPESYPKPTILWKTQLVAEPSIIEFLSQRITRSPDGALYFSNV
sf EPGPRLVLDQCD.....
ad IRGTTVLELCE.....
aa VRGSTIIELECE.....
cq IRGTTIIELECE.....
ce PGGETYQLRCD.....
tb VEGTPVSLHCA.....

```

```

dm .....
pv EYSKVVYSSFEYTC.....
tn .TEDVDGDKFKYVCYAQTAPXRDDVLLAEHKLVSLEKPKTPNDGELSLOQYVNTDITSKVG
sf .....
ad .....
aa .....
cq .....
ce .....
tb .....

```

```

          60      70      80      90
dm .....MMGSQVPSIQWVVGHLPRSELDDL.....DSNQVAEEAPSAIV.RVRS
pv .....EKDFAEYCKEFKNNLPRIEEDYDSISC..QSD.LRHEPISE...RGS
tn DVTMIYCIYGGTFLAYPDWFKENKLEAKPGD.....RITDHNRTGGKR
sf .....VFGKPAFVQWLNKNGEPVMDYEL.....ESNDILPAHPSLSA.LLTSK
ad .....IVGSPTPSVOVHVGSGQTADWEDI.....SVNVITEPTTVA.RVRS
aa .....VMGSPPTVQVHVGSGQTADWDDA.....TMNIISESSPTTMA.RVRS
cq .....VMGSPTPYVQVHVGSGQTADWEDV.....SMNVISETSPITMA.RVRS
ce .....IMSTPAATIHWFKNGKLIQGSNEL...NVEEKLNFVK..AIVDTG.IVASI
tb .....AHGVEPTPIYWFKNGEYLPQGGKLEKDRPEMELFMNIGQ..RTLQSS.NTAGO

```

```

          100      110      120      130
dm HIIDHVL.S.EARTYTCVGR.TGSKTIYASTVHPPRSS.....
pv TDDDAVPRAQAR..TVDGLTGPEA.PRDNIIVLPPSRENEDTSEKSHPGYALPPEMKDHIG
tn LLTKETLYEDQGT.YKCEVNGVGGKLTXSXKLTV.....
sf LVVRS..ANGDVYTCVAT.SALNQKTASTTVFVTDGNDEN...L.....
ad LVIDRSSRASQTYTCIGRSAGQLAMASTVVYHIDATGSRLLGNFSDL.MLLRPL..GAG
aa LVIDHSSHASESTYTCIGRSAGQALAMASTVYHVEAARAN..FTDL.M.....
cq LVIDHSSHASESTYTCIGRSAGQIIVASTTVYHVDGPRSN..FTDI.L.....
ce LTIQCPASAENSGTYSCVGYNGHQTIEVAVE...IEGEASGRS.....
tb LBIQCITVDQAGIYECVANNGITEVSAKAVLAVKENTRQNSMCRS.....

```

```

          140      150      160      170      180      190
dm RLTPKTYPGAQKPRIIYTEKTHLDLMGSNIQLP.CRVHARPRAEITWLNENKEI..VQ
pv AHDLPNTFFPGAREHLFNPDSSGNSVENGNPMTIASASLLGIPSIVFLLYKFT...
tn .....VSAPKLSQQHEKRILVKEGEDISLCKITGLPEPKVTWYTN..TKAV...
sf .LILEKLFMRPTKPIITTFYTNLQDIGSEVILP.CRVDSNSESQVWYKNNQEQLIPLFD
ad ATSPHTFMNLKNARITLHYKTLFENMGTTVVLP.CKAVGRPVPEITWLNEN.DGNVVGSLQ
aa .KPAHHLFMHLKQARVTLHYKALFEYMGSTVILP.CKTVGRPLPEVTWKNE.EGNAIGSMQ
cq .KPGHMLFMHLKQARVTLHYKALFEYMGSTVILP.CKTVGRPLPEVTWKNE.EGNAIGSMQ
ce .....NHKSAPETVFWTDSRFEMTGNVATLV.CRANQQ..VDVWVMSN..DELV..KN
tb .....GGEVPPKIVMWTDFRIEMEDASVQLFCRAGNPMPEIQWFTG.DGKRI..IN

```

```

          200      210      220      230      240
dm GHRHRVLANGDLISETKWE DMGNYKCIARNVVGKDTADTFVYVPLNEED.....
pv ..PFRALVDP.....HIRKTKKMLMNPVNDNNEFQSHDYFNFE
tn .SERAIYKDGVLKIKNAKKGDTGYGCKAENHEGDLYAETLVQVA.....
sf NSRMRVLP.SGDLVISP.LWSDMGNYSCVAKNAAYGKDTVDTFVYPLKPSKA.....
ad DFRRLTLP.TGELIITGLRWTDMGTYTCAKNAISKDEAETFFIYVVRPN.....
aa DFRFRTL.TGELVINLRLWSDMGSYTCAKNAISKDEAETFFIYVIRPN.....
cq DFRFRTL.TGELVINLRLWSDMGSYTCAKNAISKDEAETFFIYVIRPN.....
ce NDKFTVLSNGDLVINKIVWDMGTYTCAIRNFGEARQETFFIYPTAKKSSF.....
tb DNQFQILKSGDLIKRGNWKLGLYICKAINELGEDDIQTFYVPTANTTEKMEIKNSLRL

```

```

dm .....
pv TNMDFNRYNIAYESR
tn .....
sf .....
ad .....
aa .....
cq .....
ce .....
tb TSA.....

```

Reviewer #2

REVIEWER: *Insect insulin-like polypeptides binding proteins (IBPs) have been considered to be structural and functional homologues of the IGF binding proteins (IGFBP). This manuscript describes the first insect IBP structure, that of the *Drosophila Imp-L2* protein, both free and in complex with *Drosophila* insulin-like peptide 5 (DILP5) and human IGF-1. These structures identify a new, Ig-fold architecture that is quite distinct from the common structures of the IGFBPs, and which binds ILPs in a different fashion.*

The authors also suggest similar hormone binding modes in insect vectors such as mosquito, raising the possibility that these interactions may present opportunities to modulate the transmission or progression of mosquito-borne diseases such as malaria, dengue and yellow fever.

1. REVIEWER: *The Introduction is rather unfocussed. Similarly, the Discussion is too long.*

Response: The unfocussed-like character of the Introduction results inevitably from the multilevel nature of biochemical and physiological issues this research concerns. It covers the intersection of human, insect and other invertebrates complex hormonal systems. We discover some of its very new aspects and clarify a confusion in some aspects of this research field. Hence a typical introductory-like brevity is extremely difficult to achieve as it would lead to lack of clarity in further parts of this report (e.g. IGFBPs – IBPs relationship, catholic hormone binding properties of IBPs). However, we followed referee opinion and tried to cut the Introduction to much more succinct format.

Discussion was also shortened as suggested, however, referees' requests for some additional experiments inevitably required some additional cover in the Discussion.

2. REVIEWER: *p. 8: Role of electrostatic interactions in Imp-L2 – DILP5 binding: has the effect of ionic strength on binding affinity been assessed?*

Response: We duly followed this request and Imp-L2: DILP5 binding has been measured as a function of the ionic strength as 50, 150 and 300 mM NaCl, that may, roughly, mimic the variations of the osmotic pressure of the hemolymph during insect metamorphosis. Remarkably, the lowest ionic strength abolished DILP5–Imp-L2 binding, while the highest ionic strength shifted DILP5 affinity from ~8 nM into a ~ 5 pM range (new Suppl. Fig.10).

This findings correlate further with our new SEC-MALLS data on apo–Imp-L2 behaviour in similar NaCl range, as the lowest ionic strength stabilises the apo–Imp-L2 dimer that shelter the hormone binding sites, while the highest ionic strength shifts the quaternary equilibria towards free monomer with freed hormone binding surface (new Suppl. Fig.6).

These correlations are paralleled now even further by new SEC-MALLS tracing of the Imp-L2 behaviour upon binding of the hormones, which, again, shift clearly the apo-dimer into holo-monomers (new Suppl. Fig 5).

3. REVIEWER: *p. 11: in view of the failure of SPR to provide reliable Kd values, ITC could be utilised. ITC measurements would also clarify the stoichiometry.*

Response: This very valid point has been fully addressed by the new ITC measurements.

The DILP5 Imp-L2 K_d has been measured by the ITC (new Table 1, Sup. Fig 8 and 9). As a very different PEG assay with the use of the radioactive ligands was employed before (ref 25), binding of insulin and IGF-1 to ITC was remeasured by the ITC as well (New Table 1, Sup. Fig 8 and 9) to remove the methodological bias in comparison of DILP5 ITC-derived K_d to the K_d s of other hormones measured by the PEG assay. Insulin:Imp-L2 complex eluded the crystallisation (so far). The thermodynamic components of these binding data have been assessed as well (new Suppl. Fig 9). As expected DILP5 is the strongest Imp-L2 binder (8 nM), followed by IGF-1 (13.6 nM), and human insulin (135 nM).

4. REVIEWER: *Figure 4: I found the comparison between the DILP5 and human IGF-1 Imp-L2 binding modes difficult to follow from this figure.*

Response: we tried different representation of these complexes and, somehow, the views and forms depicted in Fig. 4 still were coming up as the most transparent ones. However, if necessary, we may try to produce alternative figure for the SI.

5. REVIEWER: *Has the mechanism of release of hormone from the complex with Imp-L2 been considered?*

Response: we are not aware of any reliable data or hypotheses concerning release of hormones from the IBPs. In human IGF-BPs, a proteolytic mechanism has been proposed but at this point there is no evidence for a similar mechanism in the insect systems.

6. REVIEWER: *There are numerous minor grammatical errors in the manuscript, of which the following are just a few examples:*

p. 3: and some possibly some non-signalling

p. 4: with suggested ≤ 100 -fold affinity than classical IGF-BPs,

There are also several rather convoluted sentences, which make the narrative unnecessarily difficult to follow, for example:

p. 11: Moreover, the 1-16 N-terminus of the Imp-L2 must also reallocate to accommodate the incoming A chains of the hormones, thrusting the adjacent 84-91 segment of the 70-90 Imp-L2 loop. This, likely, triggers a large sway of the whole 60-91, loop-including protein chain, initiating a tight holo-dimerisation of the Imp-L2, in an arrangement that is different from its apo-dimer.

p. 15: In summary, the structural and functional insights into apo/holo-Imp-L2 IBP system reported here transgress their importance for the insect-related regulation of the bioavailability of insulin-like hormones.

Response: We thank the referee for highlighting these technical comments. All have been addressed in the relevant places of the manuscript.

7. REVIEWER: *p. 17: the buffer conditions and temperature for SPR and SAXS experiments should be specified.*

Response: These have been provided now.

8. REVIEWER: Bibliography: all species names to be italicised.

Response: done

Reviewer #3

This work is very interesting and brings valuable and novel information on the structure of an insect insulin-like peptide binding protein (IBP). This study is the first to elucidate this structure in insects, thereby revealing significant differences with human IGF-binding proteins (IGFBPs). Here are some minor and major concerns:

1. REVIEWER: *Minor comments:*

Responses: All comments provided below concern minor technical points: all have been addressed in the relevant places of the manuscript.

Introduction:

(1.) *The authors have a strange way of showing the abbreviations. For instance: IGF-Binding Proteins (IGFBP); why are I, B and P written in italics? Perhaps it is to show where the abbreviation comes from, but this is quite obvious. Especially in this case, where also G and F are part of the abbreviation. However, the authors are not consequent in this, because a bit further (page 2) they do not only write the first character in italics, but also underline it. So, above all, try to be more consistent.*

(2.) *"This involves freely circulating IGFBPs that are referred here as ILPs Binding Proteins (IBPs)". It is not clear what you mean with 'freely circulating'. Furthermore, it is confusing to use another term (IBPs) when you are talking about the IGFBPs. Just respect existing nomenclature. So use IGFBPs when you are talking about human IGF binding proteins, and use IBP when you are talking about insulin-like peptide binding proteins (in organisms like insects).*

(3.) *"...secrete a binding protein – named Sf-IBP – that binds insulin, IGF-1 and IGF-2". I suppose you mean human (or mammalian) insulin, IGF-1 and IGF-2? Please specify this.*

(4.) *"Here, Andersen et al. (25) showed that Sf-IBP has high insulin and significant IGF-1/IGF-2 affinities (70, 170 and 370 pM, respectively (Table S1))," this sentence is not correct/clear; what do you mean here with "significant"?*

(5.) *"...cloning indicating the presence of two Ig-like C2 domains...". Explain a bit more about Ig-like domains. This is not immediately clear for the broader public of this journal.*

(6.) *last alinea: Write 3-D as 3D*

Results:

(7.) *"... consists of two b-sheets: bA'- bG-bF-bC and ..." ∅ remove the space between '- ' and 'bG'.*

(8.) *"... which joins bE strand from a different/opposite..." add '- ' between bE and strand (to be consistent).*

(9.) *"The continuous inter-domain-b-sheet (id-b-sheet, 'top-side' of the Imp-L2) is formed by a tight interface between Ig-NT bA-bB-bE and Ig-CT bAC-bG-bF-bC-bC' b-sheets that is stabilised by hydrogen bonds between parts of bE (92-99) and*

bA (152–160) strands.” Do you mean bAC (152–160) with ‘bA (152–160)’? It is confusing since the Ig–NT also has a bA sheet...

(10.) Relation of the Imp–L2 to the Ig–fold containing proteins. ‘PDB’ is an abbreviation used for the first time in the text. Please write in full.

(11.) “These mutual hormone and IBP structural rearrangements lead to closer contacts of the Imp–L2 236–239 C–terminal segment and A–chain A12–A18 linker of the A–chain α –helices of DILP–5.” Use Imp–L2 instead of IBP to be consistent and less confusing.

Discussion

(12.) “A high sequence identity (33%) between Sf IBP and Dm Imp–L2 (Fig. 6, (45)), especially in hormone–binding regions, ...”. What is the similarity of the hormone–binding regions?

Response to the above point (12): New Suppl. Figure 12 has been produced that should fully clarify this issue.

(13.) “... dengue, zika and yellow fevers (Fig. 6,).” Remove comma behind Fig. 6.

Methods

(14.) “The construct encoding Imp–L2 recombinant protein was made using polymerase chain reaction amplification of human Imp–L2 cDNA (Uniprot No: Q09024) ...” Don’t you mean Drosophila instead of human?

(15.) Where did you get DILP5, DILP2, insulin and IGF–1? Mention this in the methods section.

Figures:

(16.) Figure 1: You always speak about DILPs, but these are Drosophila ILPs. Make this scheme more general towards other insects as well. Or speak about Drosophila instead of insects.

(17.) Figure 2: “The Ig–NT and Ig–CT domains are in white and coral, respectively. NT and CT – termini of protein.” Better to write: “The Ig–NT and Ig–CT domains are in white and coral, respectively (NT and CT – termini of protein).”

(18.) Figure 2: “The white/coral colouring scheme corresponds to the Ig–NT and Ig–CT domains.” It is better to write “The white and coral colouring scheme corresponds to the Ig–NT and Ig–CT domains in a.”

2. REVIEWER: *(19.) Figure 4: the violet colour of DILP5 is not that visible. Please use another colour.*

Response: Colour scheme of Figure 4 has been changed into, we believe, a more visible scheme for the Referee and other readers.

Major Comments:

3. REVIEWER: (A.) Information about the homology of DILP5 and DILP2 is missing. Are they structurally different? Is there a difference in how they bind the Imp-L2 (crystal structure)?

Response: A new Suppl. Figure 1 has been included in the SI which presents the homology of all DILP-1-8, human insulin, human IGF-1, and IGF-2. All these hormones show very similar organisation and homology.

Our crystallization trials of Imp-L2 with human insulin and DILP-2 have, so far, been unsuccessful, hence DILP2 binding mode is still elusive. However, considering a very high homology between these two DILPs, the structural signatures of their complexes with Imp-L2 should be very similar. We aim to reconvene these efforts when we will be able to produce (fully synthesise) DILP2 and some other DILPs in our future research programme in this subject. While we have established efficient chemical methods for their production, they are not trivial and require dedicated technical expertise and time.

4. REVIEWER: (B.) How similar is DILP5 to other ILPs in other insects? And thus, are these relevant DILPs to make hypotheses about ILP:IBP interactions in other insects? Explain this in the discussion.

Response: *D. melanogaster* DILPs present the best characterised system of the ILPs. The sequences of other insects ILPs have been proposed (e.g. Mizoguchi & Okamoto, *Frontiers in Physiology* (2013), 4, 217 doi: [10.3389/fphys.2013.00217](https://doi.org/10.3389/fphys.2013.00217)) and they show very similar organization in B, A, and C domains/chains. There is a high sequence homology within these segments that suggest similar structural organisation as well. However these are proposed sequences of these ILPs in contrast to well established Dm DILPs. Hence we opted to stick here to well-proven ILPs system and do not mix it with a putative sequences of other ILP-like hormones.

5. REVIEWER: (C.) How similar is DILP5 to IGF-1? Include this in supplementary data and discuss in text.

Response: This issue has been addressed in the new Figure 2 which shows sequence alignment of these hormones and human insulin and IGF-2 too, as well as comparison of the structures of these hormones.

6. REVIEWER: (D.) Why can Imp-L2 not bind other DILPs? Discuss this in your discussion and add a figure in supplementary data to show the differences/similarities between the different DILPs.

Response: The studies of other DILPs and Imp-L2 complexes will follow. The main bottle neck of such research is not the expression and purification of the Imp-L2 (and, we think, similar IBPs) but the production and access to DILPs. The full synthesis and expression of these hormones present formidable challenges, and is probably the limiting step not only in this research but, generally, in studies of the role of these hormones in insects biochemistry and physiology. Hence we cannot answer why Imp-L2 'does not bind' other DILPs, as we did/do not have them in hand. DILP5 was the easiest, so far, to produce in quantities allowing some

systematic work here. However, we are in the process of putting together an extensive collaborative research programme and a grant application where the production of other DILPs will be undertaken.

Additionally, the dominance of weak interactions in DILP5:imp-L2 complex and lack of numerous hydrogen bonds on this hormone:IBP interface prohibit far-going speculations on possible detail of DILP2:Imp-L2 complex. Here, we express the cautious hypothesis that an ease of T-to-R transition of the N-terminus of a hormone – needed for effective binding of the B-helix on the Imp-L2 surface – may be one of important factors affecting its Imp-L2 affinity.

The similarities of DILPs, human insulin, IGF-1 and IGF-2 are presented now in the new Suppl. Fig 1.

7. REVIEWER: *(E.) You performed SPR analysis on Imp-L2:hormone interactions. Here you tested 4 hormones: insulin, IGF-1, DILP2 and DILP5. However, you only show crystal structures for IGF-1 and DILP5. How similar are these 4 hormones? And do you know if they have similar properties (structural) when binding Imp-L2? Wouldn't it be interesting to also investigate the crystal structure of insulin:Imp-L2?*

Response: Similarity of hormones is given in new Suppl. Fig 1. We tried to crystallize insulin:Imp-L2 – so far without success but there is a progress here; e.g. we will use some insulin mutants which may have higher affinity than currently used wild type insulin and its monomeric ProB28Asp mutant.

We do not know (yet) about the nature of other DILPs:imp-L2 complex formation due to lack of sufficient amounts of these hormones (see response to above p.6).

In general, we cannot solve all DILPs:Imp-L2 issue in one research paper. We believe we clarified some outstanding questions about the relationships between IGFBPs and IBPs. We have provided novel findings in this matter that allow rational design of many new research streams concerning role of ILPs and IBPs in insects biochemistry and physiology. We are also undertaking phenotypic studies with transgenic *Dm* with Imp-L2 mutated in hormone-binding region and *Dm* with Imp-L2 replaced by vector insects IBPs. This will be a major collaborative programme of significant duration.

8. REVIEWER: *(F.) IBPs of more 'primitive' (ametabolous and hemimetabolous) insects are missing in fig. 6. Include these also in your discussion.*

Response: A protein or DNA query of Imp-L2 or sIBP sequences against insects genomes using BLASTp, PSI-BLAST and PHI-BLAST was not successful in identifying clearly any other Imp-L2 homologues than those reported here in new main text Figure 7.

The sequences used in the alignment of seven different Imp-L2 homologues were chosen based on their spread across families as well as availability in the database of non-redundant sequences from BLAST. Hence the identification of other IBPs in 'primitive' insects is still 'work in progress'.

9. REVIEWER: *(G.) Results: "As a result, the N-terminal part of this bA strand (147-154: referred her as to bAN) belongs to bANbB/bB'-bE-bD b-sheet of the Ig-CT, while the C-terminal part of this strand (157-160: referred here as to bAC)*

contributes to the bAC-bG-bF-bC-bC' b-sheet of this domain." It is confusing to have A in fig. 2a and Ac and AN in fig. 2b.

Response: A new Figure 2 has been produced that, we believe, resolves all the above issues.

10. REVIEWER: *(H.) Results: "The relative flatness of the id-b-sheet surface contrasts with a shallow, concave shape of the opposite ('back') side of the Imp-L2, with its apex at the bA: bE inter-domain interface. There is an almost two b-strand width gap between the nearest Ig-NT and Ig-CT b-strands on the back side of the Imp-L2 which result in only one b-sheet thickness in the central part of this protein." This is not clear, clarify this in your figure.*

Response: A new Figure 2 has been produced which includes panel 2c that addresses this issue.

11. REVIEWER: *(I.) Results: "Oligomeric states of apo-imp-l2 in the solid state." Even after reading the discussion I'm still wondering how functionally relevant this is. Can you find these oligomers in the fruit fly itself? Perform a western blot on hemolymph or on cells overexpressing this protein to see if dimers/oligomers occur in vivo.*

Response: We deliberately limited our studies to structure-function aspects of ILPs:IBP in Dm system, without more in-depth divergence into fruit fly physiology. This is the next step in our research programme. However, we performed new SEC-MALLS and ITC analyses of Imp-L2 that provided exacting new insight into oligomeric behaviour of this protein, and shed some light on its possible association stages in vivo.

Firstly, DILP5 *K_d* to Imp-L2 has been measured by the ITC (New Table 1, Sup. Fig 8 and 9). As a very different PEG assay with the use of the radioactive ligands was employed before (ref 25), binding of insulin and IGF-1 to Imp-L2 was remeasured by the ITC as well (New Table 1, Sup. Fig 8 and 9) to remove the methodological bias in comparison of DILP5 ITC-derived *K_d* to the *K_d*s of other hormones measured by the PEG assay, The thermodynamic components of these binding data have been assessed as well (new Suppl. Fig 9). As expected DILP5 is the strongest Imp-L2 binder (8 nM), followed by IGF-1 (13.6 nM), and human insulin (135 nM).

Moreover, DILP5 - Imp-L2 binding was also measured as a function of the ionic strength (50, 150, 300 mM NaCl) that may roughly mimic the variations of the osmotic pressure of the hemolymph during insect metamorphosis. Remarkably, the lowest ionic strength abolished DILP5-Imp-L2 binding, while the highest ionic strength shifted DILP5 affinity from ~ 8 nM into a ~5 pM range (new Suppl. Fig.10).

These findings correlate further with our new SEC-MALLS data on apo-Imp-L2 behaviour in similar NaCl range, as the lowest ionic strength stabilised the apo-Imp-L2 dimer that shelter the hormone binding sites, while the highest ionic strength shifts the quaternary equilibria towards free monomer with freed hormone binding surface (new Suppl. Fig.6).

These correlations are paralleled now even further by SEC-MALLS tracing of the Imp-L2 behaviour upon binding of the hormones, which, again, shift clearly the apo-dimer into holo-monomers (new Suppl. Fig 5).

12. REVIEWER: (J.) Discussion: you talk about ‘*Anopheles aegypti*’. But as far as I know this mosquito does not exist. Do you mean *Aedes aegypti* or *Anopheles gambiae*? *Anopheles* is also written with one ‘n’ and not two.

Response: We apologise for this technical error which is now fully corrected. We provide now sequences for *Aedes aegypti*, *Anopheles darlingi*, *Culex quinquefasciatus*, *Spodoptera frugiperda*, *Caenorhabditis elegans*, and *Trichinella britori*.

13. REVIEWER: (K.) Why did you use *Imp-L2* and not an IBP that has a higher affinity for insulin like the *SfIBP*? Or why didn’t you perform this study on a more relevant insect species, like *Aedes aegypti*?

Response: Similarly to some responses above we have to stress that addressing such questions would result in one manuscript that would solve practically all issues concerning interactions of all IBPs and ILPs. We certainly plan to research *Sf-IBP* and other IBPs as well both in vitro and in vivo.

We planned and executed this work on *Dm* as this is the best investigated ILP – IBP model system with the best characterized hormones and which – most importantly – were most feasible to produce. As the initial, main aim of this project was to clarify IGFs–IBPs relationship issue the *Dm* was the best system to investigate. Moreover, the genomes of some mosquitos were released quite recently, and high homology among these IBPs and *Imp-L2* was an exciting finding at a very late stage of this research.

14. REVIEWER: (L.) “A high sequence identity (33%) between *Sf IBP* and *Dm Imp-L2* (Fig. 6, (45)), especially in hormone-binding regions, suggest that their overall folds and hormone binding modes are very similar, hence may be representative of other members of insect IBP family. The differences in specificity of hormone binding (e.g. insulin vs IGF-1) between *Sf IBP* and *Imp-L2* (Table 1) would then result from subtle side chain differences in some specific regions of these IBPs, rather than from their different tertiary structures.” I think this hypothesis is too strongly formulated. Tone down. Because you don’t know this for sure based on the data you present in this paper.

Response: This – and similar sentences – have been toned down as requested. Also, a new Suppl. Figure 12 supports further our argument of possible structural similarities of hormone binding modes by the IBPs. There is a remarkable conservation within IBPs C-terminal domains that are likely involved in hormone binding, with emerging sequence signatures for IBPs that should help in the future in search for and classification of these proteins.

14. REVIEWER: (M.) “The *Imp-L2* structures expand the already abundant superfamily of the Ig-fold – the most coded metazoan module (54) – onto their new structural variations and functional application. The fusion of two Ig-domains in *Imp-L2*, and, likely, in the other insects IBPs, is different from similar motives found in even structurally very related human muscle protein titin M10 domain, and its complexes that rely on extensive Ig-Ig domain interactions. Their ‘head-to-tail’

arrangement in M10 complexes, which may be dictated by the directionality of these filaments in the muscle M-band, is very different from the Imp-L2 ‘mirror-image’-like Ig-NT:Ig-CT fold.” Similar as previous comment. Tone down.

Response: These statements have been softened as requested.

15. REVIEWER: (N.) *“Moreover, insect IBPs show capability of enforcing an allosteric effect on insulin-like hormones, inducing ...” “Speaking about insect IBPs is far too general. Tone this down. It may not be the case for all insect IBPs.*

Response: Now modified into a more moderate statement as requested.

REVIEWERS' COMMENTS:

Reviewer #1 (Remarks to the Author):

The authors have provided detailed and thoughtful responses to all the Reviewers' comments, and have substantially modified and significantly improved their manuscript. Interesting new data on ligand affinities and influence of ionic strength have been added, and presentational changes have been made, which address key major concerns of Reviewers 1, 2 and 3. Unfortunately unavailability of reagents and fully annotated genomic sequences stands in the way of further revision, so this is not something that can be insisted on. It is clear that the pathophysiological implications are unlikely to be resolved by do-able experiments in the near future. In the meantime the revised manuscript provides a balanced account of both molecular and physiological aspects of the data.

Minor Point: In presenting previously published binding data (now Table 1 in main manuscript) the authors still cite ref 27 when this should be ref 25. It might be a good idea to check all ref citations to make sure other errors have not crept in during manuscript revisions.

Signed: K Siddle

Reviewer #3 (Remarks to the Author):

Minor comment:

You introduced a new suppl. Figure 12. It would help the reader if you state what the numbers in the right corner of the graph mean. Which regions are they in the sequence? Add this to the figure legend.

REVIEWER #1 (REMARKS TO THE AUTHOR):

The authors have provided detailed and thoughtful responses to all the Reviewers' comments, and have substantially modified and significantly improved their manuscript. Interesting new data on ligand affinities and influence of ionic strength have been added, and presentational changes have been made, which address key major concerns of Reviewers 1, 2 and 3. Unfortunately unavailability of reagents and fully annotated genomic sequences stands in the way of further revision, so this is not something that can be insisted on. It is clear that the pathophysiological implications are unlikely to be resolved by do-able experiments in the near future. In the meantime the revised manuscript provides a balanced account of both molecular and physiological aspects of the data.

RESPONSE: We are grateful to the Reviewer for considering our previous responses/additional data/work as, altogether, "a balanced account of both molecular and physiological aspects of the data". We fully agree with the Reviewer that there is plenty to be done to get any insight into relevant and pathophysiological implications of the IBP system and its (if any) interplay with human physiology. We only aimed here to open new streams and venues of research on the IBPs providing some foundation for the future work in this field, which, we hope will be followed/joined by other researchers interested in these fascinating, but experimentally most challenging, problems.

Minor Point: In presenting previously published binding data (now Table 1 in main manuscript) the authors still cite ref 27 when this should be ref 25. It might be a good idea to check all ref citations to make sure other errors have not crept in during manuscript revisions.

Signed: K Siddle

RESPONSE: The reference 25 in Table 1 has been used/corrected as requested by the Reviewer #1. All references have been carefully checked as suggested.

REVIEWER #3 (REMARKS TO THE AUTHOR):

Minor comment:

You introduced a new suppl. Figure 12. It would help the reader if you state what the numbers in the right corner of the graph mean. Which regions are they in the sequence? Add this to the figure legend.

RESPONSE: The insert in the new Supplementary Figure 12, i.e. the colour coding/numbering of the compared segments of the Imp-L2/IBPs sequences has been explained now in the figure legend as requested.